# Novel genetic loci affecting facial shape variation in humans

Ziyi Xiong[1,2,3†], Gabriela Dankova[1†], Laurence J Howe[4], Myoung Keun Lee[5], Pirro G Hysi[6], Markus A de Jong[1,7,8§], Gu Zhu[9], Kaustubh Adhikari[10], Dan Li[11,12,13], Yi Li[3], Bo Pan[14], Eleanor Feingold[5], Mary L Marazita[5,15], John R Shaffer[5,15], Kerrie McAloney[9], Shu-Hua Xu[11,12,13,16,17], Li Jin[11,12,13,18,19], Sijia Wang[11,12,13,17], Femke MS de Vrij[20], Bas Lendemeijer[20], Stephen Richmond[21], Alexei Zhurov[21], Sarah Lewis[4], Gemma C Sharp[4,22], Lavinia Paternoster[4], Holly Thompson[4], Rolando Gonzalez-Jose[23], Maria Catira Bortolini[24], Samuel Canizales-Quinteros[25], Carla Gallo[26], Giovanni Poletti[26], Gabriel Bedoya[27], Francisco Rothhammer[28], André G Uitterlinden[2,29], M Arfan Ikram[2], Eppo Wolvius[7], Steven A Kushner[20], Tamar EC Nijsten[30], Robert-Jan TS Palstra[31], Stefan Boehringer[8], Sarah E Medland[9], Kun Tang[11,12,13], Andres Ruiz-Linares[18,19,32], Nicholas G Martin[9], Timothy D Spector[6‡], Evie Stergiakouli[4,22‡], Seth M Weinberg[5,15,33‡], Fan Liu[1,3‡*], Manfred Kayser[1‡*], On behalf of the International Visible Trait Genetics (VisiGen) Consortium

[1]Department of Genetic Identification, Erasmus MC University Medical Center Rotterdam, Rotterdam, Netherlands; [2]Department of Epidemiology, Erasmus MC University Medical Center Rotterdam, Rotterdam, Netherlands; [3]CAS Key Laboratory of Genomic and Precision Medicine, Beijing Institute of Genomics, University of Chinese Academy of Sciences (CAS), Beijing, China; [4]Medical Research Council Integrative Epidemiology Unit, Population Health Sciences, University of Bristol, Bristol, United Kingdom; [5]Center for Craniofacial and Dental Genetics, Department of Oral Biology, University of Pittsburgh, Pittsburgh, United States; [6]Department of Twin Research and Genetic Epidemiology, King's College London, London, United Kingdom; [7]Department of Oral & Maxillofacial Surgery, Special Dental Care, and Orthodontics, Erasmus MC University Medical Center Rotterdam, Rotterdam, Netherlands; [8]Department of Biomedical Data Sciences, Leiden University Medical Center, Leiden, Netherlands; [9]QIMR Berghofer Medical Research Institute, Brisbane, Australia; [10]Department of Genetics, Evolution, and Environment, University College London, London, United Kingdom; [11]CAS Key Laboratory of Computational Biology, Chinese Academy of Sciences (CAS), Shanghai, China; [12]CAS-MPG Partner Institute for Computational Biology (PICB), Chinese Academy of Sciences (CAS), Shanghai, China; [13]Shanghai Institute of Nutrition and Health, Shanghai Institutes for Biological Sciences, Chinese Academy of Sciences (CAS), Shanghai, China; [14]Department of Auricular Reconstruction, Plastic Surgery Hospital, Beijing, China; [15]Department of Human Genetics, University of Pittsburgh, Pittsburgh, United States; [16]School of Life Science and Technology, ShanghaiTech University, Shanghai, China; [17]Center for Excellence in Animal Evolution and Genetics, Chinese Academy of Sciences, Kunming, China; [18]State Key Laboratory of Genetic Engineering, School of Life Sciences, Fudan University, Shanghai, China; [19]Ministry of Education Key Laboratory of Contemporary Anthropology, School of Life Sciences, Fudan University, Shanghai, China; [20]Department of Psychiatry, Erasmus MC University Medical Center Rotterdam, Rotterdam, Netherlands; [21]Applied Clinical Research and Public Health,

**\*For correspondence:**
liufan@big.ac.cn (FL);
m.kayser@erasmusmc.nl (MK)

[†]These authors also contributed equally to this work
[‡]These authors also contributed equally to this work

**Present address:** [§]Department of Computer Science, VU University Amsterdam, Amsterdam, Netherlands

**Competing interests:** The authors declare that no competing interests exist.

University Dental School, Cardiff University, Cardiff, United Kingdom; [22]School of Oral and Dental Sciences, University of Bristol, Bristol, United Kingdom; [23]Instituto Patagonico de Ciencias Sociales y Humanas, CENPAT-CONICET, Puerto Madryn, Argentina; [24]Departamento de Genetica, Universidade Federal do Rio Grande do Sul, Porto Alegre, Brazil; [25]UNAM-Instituto Nacional de Medicina Genomica, Facultad de Quımica, Unidad de Genomica de Poblaciones Aplicada a la Salud, Mexico City, Mexico; [26]Laboratorios de Investigacion y Desarrollo, Facultad de Ciencias y Filosofıa, Universidad Peruana Cayetano Heredia, Lima, Peru; [27]GENMOL (Genetica Molecular), Universidad de Antioquia, Medellın, Colombia; [28]Instituto de Alta Investigacion, Universidad de Tarapaca, Arica, Chile; [29]Department of Internal Medicine, Erasmus MC University Medical Center Rotterdam, Rotterdam, Netherlands; [30]Department of Dermatology, Erasmus MC University Medical Center Rotterdam, Rotterdam, Netherlands; [31]Department of Biochemistry, Erasmus MC University Medical Center Rotterdam, Rotterdam, Netherlands; [32]Aix-Marseille Université, CNRS, EFS, ADES, Marseille, France; [33]Department of Anthropology, University of Pittsburgh, Pittsburgh, United States

**Abstract** The human face represents a combined set of highly heritable phenotypes, but knowledge on its genetic architecture remains limited, despite the relevance for various fields. A series of genome-wide association studies on 78 facial shape phenotypes quantified from 3-dimensional facial images of 10,115 Europeans identified 24 genetic loci reaching study-wide suggestive association ($p < 5 \times 10^{-8}$), among which 17 were previously unreported. A follow-up multi-ethnic study in additional 7917 individuals confirmed 10 loci including six unreported ones ($p_{adjusted} < 2.1 \times 10^{-3}$). A global map of derived polygenic face scores assembled facial features in major continental groups consistent with anthropological knowledge. Analyses of epigenomic datasets from cranial neural crest cells revealed abundant *cis*-regulatory activities at the face-associated genetic loci. Luciferase reporter assays in neural crest progenitor cells highlighted enhancer activities of several face-associated DNA variants. These results substantially advance our understanding of the genetic basis underlying human facial variation and provide candidates for future in-vivo functional studies.

## Introduction

The human face represents a multi-dimensional set of correlated, mostly symmetric, complex phenotypes with high heritability. This is illustrated by a large degree of facial similarity between monozygotic twins and, albeit less so, between other relatives (*Djordjevic et al., 2016*), stable facial features within and differences between major human populations, and the enormous diversity among unrelated persons almost at the level of human individualization. Understanding the genetic basis of human facial variation has important implications for several disciplines of fundamental and applied sciences, including human genetics, developmental biology, evolutionary biology, medical genetics, and forensics. However, current knowledge on the specific genes influencing human facial appearance and the underlying molecular mechanisms forming facial morphology is limited.

Over recent years, nine separate genome-wide association studies (GWASs) have each highlighted several but largely non-overlapping genetic loci associated with facial shape phenotypes (*Adhikari et al., 2016*; *Cha et al., 2018*; *Claes et al., 2018*; *Cole et al., 2016*; *Lee et al., 2017*; *Liu et al., 2012*; *Paternoster et al., 2012*; *Pickrell et al., 2016*; *Shaffer et al., 2016*). The overlapping loci from independent facial GWASs, thus currently representing the most established genetic findings, include DNA variants in or close to 10 genes that is *CACNA2D3* (*Paternoster et al., 2012*; *Pickrell et al., 2016*), *DCHS2* (*Adhikari et al., 2016*; *Claes et al., 2018*), *EPHB3* (*Claes et al., 2018*; *Pickrell et al., 2016*), *HOXD* cluster (*Claes et al., 2018*; *Pickrell et al., 2016*), *PAX1* (*Adhikari et al., 2016*; *Shaffer et al., 2016*), *PAX3* (*Adhikari et al., 2016*; *Claes et al., 2018*; *Liu et al., 2012*; *Paternoster et al., 2012*; *Pickrell et al., 2016*), *PKDCC* (*Claes et al., 2018*; *Pickrell et al., 2016*),

*SOX9* (*Cha et al., 2018*; *Claes et al., 2018*; *Pickrell et al., 2016*), *SUPT3H* (*Adhikari et al., 2016*; *Claes et al., 2018*; *Pickrell et al., 2016*), and *TBX15* (*Claes et al., 2018*; *Pickrell et al., 2016*), which were identified mostly in Europeans.

Indicated by the small effect size of the previously identified face-associated DNA variants, together with the high heritability of facial phenotypes, the true number of genes that influence polygenetic facial traits is likely to be much larger than currently known. Despite the previous work, our understanding of the complex genetic architecture of human facial morphology thus remains largely incomplete. This emphasizes on the need for well-powered GWASs that can be achieved through large collaborative efforts, which has been recently demonstrated for appearance traits involving pigmentation variation (*Hysi et al., 2018*; *Visconti et al., 2018*), but is currently missing for the human face.

A most recent study (*Claes et al., 2018*) suggested that variants at genetic loci implicated in human facial shape are involved in the epigenetic regulation of human neural crest cells, suggesting that at least some of the functional variants may reside within regulatory elements such as enhancers. Although this is in line with evidence for other appearance phenotypes such as human pigmentation traits, where experimental studies have proven enhancer effects of pigmentation-associated DNA variants (*Visser et al., 2012*; *Visser et al., 2014*; *Visser et al., 2015*), experimental evidence for understanding the functional basis of DNA variants for which facial phenotype associations have been previously established is missing as of yet.

Led by the International Visible Trait Genetics (VisiGen) Consortium and together with its study partners, the current study represents a collaborative effort to identify novel genetic variants involved in human facial variation in the largest set of multi-ethnic samples available thus far. Moreover, it demonstrates the functional consequences of identified face-associated DNA variants based on in-silico work and in-vitro experimental evidence.

## Results

### Facial phenotypes

The current study included seven cohorts, totaling 18,032 individuals. Demographic characteristics and phenotyping details for all cohorts are provided in Materials and methods and *Supplementary file 1* - Table 1. A variety of different phenotyping methods are available for 3D facial image data, where some approaches (*Claes et al., 2018*) require access to the underlying raw image datasets, which often cannot be shared between different research groups. In order to maximize sample size via a collaborative study, we focused on linear distance measures of facial features that can be accurately derived from facial image datasets. In our cohorts with 3D facial surface images, we focused on distances between a common set of thirteen well-defined facial landmarks (*Figure 1*). In the Rotterdam Study (RS) and the TwinsUK study (TwinsUK), we used the same ensemble method for automated 3D face landmarking that was specifically designed for large cohort studies with the flexibility of landmark choice and high robustness in dealing with various image qualities (*de Jong et al., 2018*; *de Jong et al., 2016*). Facial landmarking in Avon Longitudinal Study of Parents and their Children (ALSPAC) and Pittsburgh 3D Facial Norms study (PITT) was done manually. Generalized Procrustes Analysis (GPA) was used to remove affine variations due to shifting, rotation and scaling (*Figure 1—figure supplement 1*). A total of 78 Euclidean distances involving all combinatorial pairs of the 13 facial landmarks were considered as facial phenotypes.

Detailed phenotype analyses were conducted in RS. In RS, almost all (72 out of 78, 92%,) of the facial phenotypes followed the normal distribution as formally investigated by applying the Shapiro-Wilk normality test and obtaining non-significant results (*Supplementary file 1* - Table 2). For only six phenotypes, this test revealed significant results; however, their histogram looked largely normal. A highly significant sex effect (min p=2.1 $\times$ 10$^{-151}$) and aging effect (min p=8.0 $\times$ 10$^{-44}$) were noted, together explained up to 21% of the variance per facial phenotype (e.g. between alares and Ls; *Figure 1—figure supplement 2A–2C*, *Supplementary file 1* - Table 3). Intra-organ facial phenotypes (e.g. within nose or within eyes) showed on average higher correlations than inter-organ ones (e.g. between nose and eyes), and symmetric facial phenotypes showed higher correlations than non-symmetric ones. Genetic correlations (*Zhou and Stephens, 2014*) estimated using genome-wide data were similar to correlations obtained directly from phenotype data, with on average

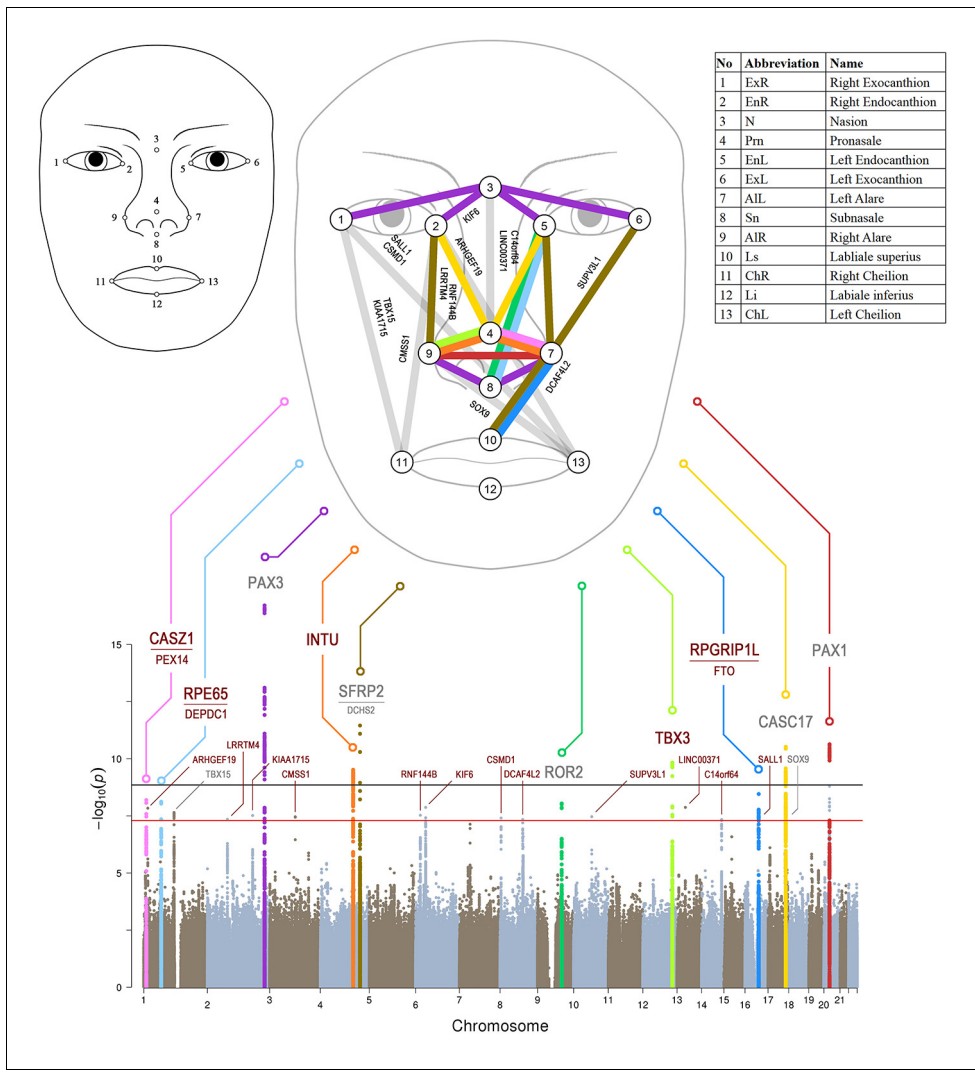

**Figure 1.** Manhattan plot from meta-analysis discovery GWAS (N = 10,115 Europeans) for 78 Euclidean distances between 13 facial landmarks. Results from meta-analysis of four GWASs in Europeans (N = 10,115), which were separately conducted in RS, TwinsUK, ALSPAC, and PITT for 78 facial shape phenotypes that are displayed in a signal 'composite' Manhattan plot at the bottom of the figure. All loci consisting of SNPs reaching study-wide suggestive significance (p<5 × 10$^{-8}$, red line) were nominated according to the nearby candidate genes in the associated loci, and the top 10 loci were highlighted in colors. The black line indicates study-wide significance after multiple trait correction (p<1.2×10$^{-9}$). Novel and previously established face-associated genetic loci were differentiated using colored and gray text, respectively. Annotation of the 13 facial landmarks used to derive the 78 facial phenotypes and the associated facial phenotypes are illustrated in the upper part of the figure.

The online version of this article includes the following figure supplement(s) for figure 1:

**Figure supplement 1.** An example of GPA, facial landmarks obtained from 3dMD images in 3193 participants from the discovery cohorts RS.

**Figure supplement 2.** Phenotype characteristics of 78 facial traits.

**Figure supplement 3.** Phenotypic and genetic correlation matrix between all facial phenotypes in cohort RS.

**Figure supplement 4.** Quantile-Quantile plots and genomic inflation factors for 22 significant associated facial traits.

**Figure supplement 5.** Power analysis.

---

higher absolute values seen based on genotypes (*Figure 1—figure supplement 3*, *Supplementary file 1* - Tables 4 and 5).

Unsupervised hierarchical clustering analysis identified four distinct clusters of facial shape phenotypes, for example many distances related to pronasale were clustered together (*Figure 1—figure*

*supplement 2D*). Twin heritability of all facial phenotypes was estimated in TwinsUK as $h^2$ mean(sd) =0.52 (0.15), max = 0.86 and in Queensland Institute of Medical Research study (QIMR) as $h^2$ mean (sd)=0.55 (0.19), max = 0.91 (*Supplementary file 1* - Table 6). The most heritable features were linked to the central face (i.e., the outer- and inter- orbital distances, the nose breadth, and the distance between the subnasion and the inner eye corners; *Figure 1—figure supplement 2E and F*). These twin-derived heritability estimates were generally higher than those obtained from genome-wide SNPs in the same cohorts (*Supplementary file 1* - Table 6). These results were overall consistent with expectations and suggest that the phenotype correlation within the human face may be explained in part by sets of shared genetic components.

## GWASs and replications

We conducted a series of GWASs and meta-analyses to test the genetic association of 7,029,494 autosomal SNPs with 78 face phenotypes in RS, TwinsUK, ALSPAC, and PITT, totaling 10,115 subjects of European descent, used as the discovery dataset. Inflation factors were in an acceptable range in all meta-analyses (average $\lambda$ = 1.015, sd = 0.006; *Figure 1—figure supplement 4*), and were thus not further considered. A matrix spectral decomposition analysis estimated the effective number of independent facial traits as 43, which corresponds to a Bonferroni corrected study-wide significant threshold of $1.2 \times 10^{-9}$. A power analysis showed that under this threshold, our discovery data had over 90% power to detect an allelic effect explaining 0.54% variance for individual phenotypes (*Figure 1—figure supplement 5A*). In the discovery GWAS meta-analysis, a total of 221 SNPs from six distinct loci showed study-wide significant association (p<$1.2 \times 10^{-9}$) with facial phenotypes, among which two have not been previously reported: 4q28.1 *INTU* (top SNP rs1250495 p=$3.03 \times 10^{-10}$), and 12q24.21 *TBX3* (rs1863716 p=$1.47 \times 10^{-10}$). The other four have been described in previous GWASs of facial variation, including 2q36.1 *PAX3*, 4q31.3 *SFRP2*, 17q24.3 *CASC17*, and 20p11.22 *PAX1* (*Figure 1* and *Supplementary file 1* - Table 7) (*Liu et al., 2012*; *Paternoster et al., 2012*; *Adhikari et al., 2016*; *Shaffer et al., 2016*; *Claes et al., 2018*). Considering the traditional genome-wide significant threshold of $5 \times 10^{-8}$ as our study-wide suggestive significance threshold for replication studies, the power analysis showed that under this threshold, our discovery data had over 90% power to detect an allelic effect explaining 0.43% variance for individual phenotypes (*Figure 1—figure supplement 5B*). In addition to the study-wide significant ones, 273 SNPs from 18 distinct loci passed the $5 \times 10^{-8}$ threshold (*Supplementary file 1* - Table 7), of which 15 were novel and 3 (1p12 *TBX15*, 9q22.31 *ROR2* and 17q24.3 *SOX9*) have been reported in previous studies (*Cha et al., 2018*; *Claes et al., 2018*; *Pickrell et al., 2016*). Thus, in total, we identified 24 face-associated genetic loci, of which 17 were not reported in previous face GWASs, that we followed-up via a multi-ethnic replication analysis (see below). In general, no significant heterogeneity was observed for the 24 top SNPs of the study-wide significant and suggestive associated loci, that is only one SNP (rs3403289 at 2q36.1 *PAX3*) showed nominally significant (Cochran's Q test p=0.04, *Table 1*) heterogeneity between the discovery cohorts, but was not significant after multiple testing correction.

The association signals at the 24 genetic loci involved multiple facial phenotypes clustered to the central region of the face above the upper lip (*Figure 1*), and closely resembled the twins-based face heritability map (*Figure 1—figure supplement 2E and F*). All seven previously reported loci showed largely consistent allele effects on the same or similar sets of facial phenotypes as described in previous GWASs (*Supplementary file 1* - Table 8). Four (2q36.1 *PAX3*, 9q22.31 *ROR2*, 17q24.3 *CASC17*, and 20p11.22 *PAX1*) out of the 24 face-associated loci have been noted in the GWAS catalog for face morphology or related phenotypes that is nose size, chin dimples, monobrow thickness, and male pattern baldness (*Supplementary file 1* - Table 9). Five out of the 24 loci showed genome-wide significant association (p<5e-8) with non-facial phenotypes and/or diseases in the GeneATLAS database (*Canela-Xandri et al., 2018*) (*Supplementary file 1* - Table 10), suggesting pleiotropic effects. Six out of the 24 loci showed a high degree of diversity among the major continental groups in terms of both allele frequency differences (>0.2) and Fst values (empirical p<0.05), including 1p31.2 *RPE65*, 2q36.1 *PAX3*, 4q28.1 *INTU*, 16q12.1 *SALL1*, 16q12.2 *RPGRIP1L* and 17q24.3 *CASC17* (*Supplementary file 1* - Tables 11 and 12). Three out of the 24 face-associated loci showed signals of positive selection in terms of the integrated haplotype scores (iHS, empirical p<0.01), including *RPE65*, *RPGRIP1L* and *CASC17* (*Supplementary file 1* - Table 12). None of the 24 loci overlapped with the previously reported signals of singleton density score (SDS, empirical

**Table 1.** SNPs significantly associated ($p<5\times10^{-8}$) with facial shape phenotypes from European discovery GWAS meta-analysis (RS, TwinksUK, ALSPAC, and PITT), and their multi-ethnic replication (UYG, CANDELA and QIMR).

| Region | SNP | Nearest Gene | EA | OA | Trait | Discovery meta-analysis (N = 10,115) | | | Replication (N = 7,917) |
|--------|-----|--------------|----|----|-------|------|---|---|---|
| | | | | | | Beta | P | Q | Com. P |
| Novel face-associated loci | | | | | | | | | |
| 1p36.22 | rs143353512 | *CASZ1* | A | G | Prn-AlL | −0.29 | $6.44 \times 10^{-9}$ | 0.73 | **0.0001** |
| 1p36.13 | rs200243292 | *ARHGEF19* | I | T | EnR-ChL | −0.11 | $1.46 \times 10^{-8}$ | 0.54 | 0.0640 |
| 1p31.2 | rs77142479 | *RPE65* | C | A | EnL-Sn | −0.08 | $7.65 \times 10^{-9}$ | 0.90 | 0.0121 |
| 2p12 | rs10202675 | *LRRTM4* | T | C | EnR-AlR | −0.26 | $4.43 \times 10^{-8}$ | 0.40 | 0.0226 |
| 2q31.1 | rs2884836 | *KIAA1715* | T | C | ExR-ChR | −0.09 | $3.04 \times 10^{-8}$ | 0.52 | 0.1096 |
| 3q12.1 | rs113663609 | *CMSS1* | A | G | EnR-ChR | −0.12 | $3.46 \times 10^{-8}$ | 0.96 | 0.0782 |
| 4q28.1 | rs12504954 | *INTU* | A | G | Prn-AlL | −0.09 | $3.03 \times 10^{-10}$ | 0.52 | **0.0015** |
| 6p22.3 | rs2225718 | *RNF144B* | C | T | EnR-AlR | −0.09 | $2.97 \times 10^{-8}$ | 0.60 | 0.0071 |
| 6p21.2 | rs7738892 | *KIF6* | T | C | EnR-N | 0.11 | $1.34 \times 10^{-8}$ | 0.09 | **0.0018** |
| 8p23.2 | rs1700048 | *CSMD1* | C | A | ExR-ChL | −0.22 | $3.94 \times 10^{-8}$ | 0.73 | 0.1131 |
| 8q21.3 | rs9642796 | *DCAF4L2* | A | G | AlL-Ls | −0.11 | $4.52 \times 10^{-8}$ | 0.07 | 0.3090 |
| 10q22.1 | rs201719697 | *SUPV3L1* | A | D | ExL-AlL | −0.28 | $3.42 \times 10^{-8}$ | 0.66 | NA |
| 12q24.21 | rs1863716 | *TBX3* | A | C | Prn-AlR | −0.09 | $1.47 \times 10^{-10}$ | 0.53 | $2.45 \times 10^{-5}$ |
| 13q14.3 | rs7325564 | *LINC00371* | T | C | N-Prn | 0.09 | $1.34 \times 10^{-8}$ | 0.16 | 0.3519 |
| 14q32.2 | rs1989285 | *C14orf64* | G | C | N-Prn | −0.10 | $4.54 \times 10^{-8}$ | 0.82 | **0.0019** |
| 16q12.1 | rs16949899 | *SALL1* | T | G | ExR-ChL | −0.13 | $3.35 \times 10^{-8}$ | 0.76 | 0.8912 |
| 16q12.2 | rs7404301 | *RPGRIP1L* | G | A | AlL-Ls | −0.09 | $3.49 \times 10^{-9}$ | 0.43 | $2.87 \times 10^{-7}$ |
| Previously reported face-associated loci | | | | | | | | | |
| 1p12 | rs1229119 | *TBX15* | T | C | ExR-ChR | 0.08 | $2.25 \times 10^{-8}$ | 0.26 | 0.1219 |
| 2q36.1 | rs34032897 | *PAX3* | G | A | EnR-N | 0.14 | $1.96 \times 10^{-17}$ | 0.04 | **0.0020** |
| 4q31.3 | rs6535972 | *SFRP2* | C | G | EnL-AlL | 0.12 | $3.51 \times 10^{-12}$ | 0.12 | $5.89 \times 10^{-12}$ |
| 9q22.31 | rs2230578 | *ROR2* | C | T | EnL-Sn | 0.09 | $9.02 \times 10^{-9}$ | 0.66 | $1.09 \times 10^{-6}$ |
| 17q24.3 | rs8077906 | *CASC17* | A | G | Prn-EnL | −0.09 | $3.00 \times 10^{-11}$ | 0.24 | 0.0505 |
| 17q24.3 | rs35473710 | *SOX9* | G | A | AlR-ChL | 0.19 | $3.95 \times 10^{-8}$ | 0.37 | 0.2634 |
| 20p11.22 | rs4813454 | *PAX1* | T | C | AlL-AlR | −0.10 | $2.32 \times 10^{-11}$ | 0.58 | **0.0002** |

Except rs2230578 (3'-UTR of *ROR2*) and rs7738892 (a synonymous variant of *KIF6*), all SNPs are intronic or intergenic. Gene symbols in bold indicate successful replication after Bonferroni correction of 24 loci ($p<2.1\times10^{-3}$). Meta P in bold indicates study-wide significance in the discovery analysis after multiple trait correction ($p<1.2\times10^{-9}$). Replication P in bold means significant after Bonferroni correction for multiple tests ($p<2.1\times10^{-3}$). Q: p-value from the Cochran's Q test for testing heterogeneity between discovery cohorts. EA and OA, the effect allele and the another allele. Com. P: p-values from a combined test of dependent tests. NA, not available.

$p<0.01$) in Europeans (*Field et al., 2016*), which may indicate that the observed selection signals are unlikely the consequences from the recent selective pressures detectable via SDS analysis.

We selected the 24 top-associated SNPs (*Table 1*), one SNP per for each of the identified 24 genetic loci passing the study-wide suggestive significance threshold ($p<5\times10^{-8}$), for replication studies in a total of 7917 multi-ethnic subjects from three additional cohorts that is the Consortium for the Analysis of the Diversity and Evolution of Latin America (CANDELA) study involving Latin Americans known to be of admixed Native American and European ancestry, the Xinjiang Uyghur (UYG) study from China known to be of admixed East Asian and European ancestry, and the Queensland Institute of Medical Research (QIMR) study from Australia involving Europeans. We used a 'combined test of dependent tests' (*Kost and McDermott, 2002*) to account for the incompatibility among facial traits from the three replication cohorts, which tests for H0: no association for all facial phenotypes in all replication cohorts vs. H1: associated with at least one facial phenotypes

in at least one replication cohorts. After Bonferroni correction of multiple testing of 24 SNPs, significant evidence of replication ($p<2.1\times10^{-3}$) was obtained for 10 SNPs, among which six were novel loci: 1p36.22 *CASZ1*, 4q28.1 *INTU*, 6p21.2 *KIF6*, 12q24.21 *TBX3*, 14q32.2 *C14orf64*, and 16q12.2 *RPGRIP1L* (*Figure 2*), and four have been described in previous face GWASs: 2q36.1 *PAX3*, 4q31.3 *SFRP2*, 9q22.31 *ROR2*, and 20p11.22 *PAX1* (*Adhikari et al., 2016*; *Claes et al., 2018*; *Paternoster et al., 2012*; *Pickrell et al., 2016*; *Shaffer et al., 2016*).

The allelic effects of the replicated SNPs were highly consistent across all participating cohorts, while the individual cohorts were underpowered (*Figure 2—figure supplements 1–6*), further emphasizing on the merit of our collaborative study. The associated phenotypes of the replicated SNPs demonstrated a clear pattern of facial symmetry that is the same association was observed on both sides of the face, while such a symmetric pattern was less clear for most phenotypes involved in the non-replicated loci (*Figure 2—figure supplements 7–17*). Several novel replicated loci showed effects on the similar sets of facial shape phenotypes; for example, 1p36.22 *CASZ1* and 4q28.1 *INTU* (*Figure 2A and C*) both affected the distances between pronasion and alares. Moreover, several novel replicated loci affected phenotypes in multiple facial regions for example 4q32.1 *C14orf64* and 12q24.1 *TBX3* (*Figure 2D and E*) both affected many distances among eyes, alares and mouth. It should be noted however, that the discovery dataset is of European ancestry while the replication dataset is multi-ethnic and the facial phenotypes could not be obtained in the same way as in the discovery set; hence, the replication results were rather conservative, that is the possibility of false negatives in this replication study cannot be excluded. This was evident from the results of the replication attempts of the seven known loci (1p12 *TBX15*, 2q36.1 *PAX3*, 4q31.3 *SFRP2*, 9q22.31 *ROR2*, 17q24.3 *CASC17*, 17q24.3 *SOX9*, 20p11.22 *PAX1*) (*Figure 2—figure supplements 18–24*). Out of these 7, only four were replicated (*PAX3*, *SFRP2*, *ROR2* and *PAX1*, *Table 1*). For this reason, we considered all of the 494 SNPs from the 24 loci passing the study-wide suggestive threshold in the subsequent polygenic score analyses and in the functional annotation analyses.

## Integration with previous literature knowledge

In our discovery GWAS dataset, we also investigated 122 face-associated SNPs reported in previously published facial GWASs (*Adhikari et al., 2016*; *Cha et al., 2018*; *Claes et al., 2018*; *Cole et al., 2016*; *Lee et al., 2017*; *Liu et al., 2012*; *Paternoster et al., 2012*; *Pickrell et al., 2016*; *Shaffer et al., 2016*). Out of those, a total of 44 (36%) SNPs showed significant association on either the same set of facial phenotypes or at least one facial phenotypes in a combined test, after Bonferroni correction for the number of SNPs (adjusted $p<0.05$) (*Supplementary file 1* - Table 8). Notably, none of the 5 SNPs reported in a previous African GWAS (*Cole et al., 2016*) was replicated in our European dataset. This is unlikely explained by the allele frequency differences between Sub-Saharan African (AFR) and Europeans (EUR) (absolute frequency differences between AFR and EUR were less than 0.1 in the 1000 Genomes project) but more likely due to different phenotypes, that is in the previous study these loci were primarily associated with facial size, which was not tested on our study.

We also investigated the potential links between facial phenotypes and 60 SNPs in 38 loci previously implicated in non-syndromic cleft lip and palate (NSCL/P) phenotype, which is linked with normal variation of facial morphology as suggested previously (*Beaty et al., 2010*; *Beaty et al., 2011*; *Birnbaum et al., 2009*; *Grant et al., 2009*; *Leslie et al., 2017*; *Ludwig et al., 2017*; *Ludwig et al., 2012*; *Mangold et al., 2010*; *Sun et al., 2015*; *Yu et al., 2017*). Among these, 17 SNPs in 13 loci showed significant face-associations in the discovery cohorts after Bonferroni correction for the number of SNPs and phenotypes, and many (11 SNPs in eight loci) involving cleft related facial landmarks as expected, for example subnasale, labliale superius, and labiale inferius (*Supplementary file 1* - Table 13).

## Multivariable model fitting and polygenic face scores

In the RS cohort, a multiple-regression conditioning on the effects of the 24 lead SNPs identified 31 SNPs showing significant (adjusted $p<0.05$) independent effects on sex- and age-adjusted face phenotypes (*Supplementary file 1* - Table 14). These 31 SNPs individually explain less than 1%, and together explain up to 4.62%, of the variance for individual phenotypes (*Figure 3A and B*). The top-explained phenotypes clustered to the central region of the face, that is the distances between

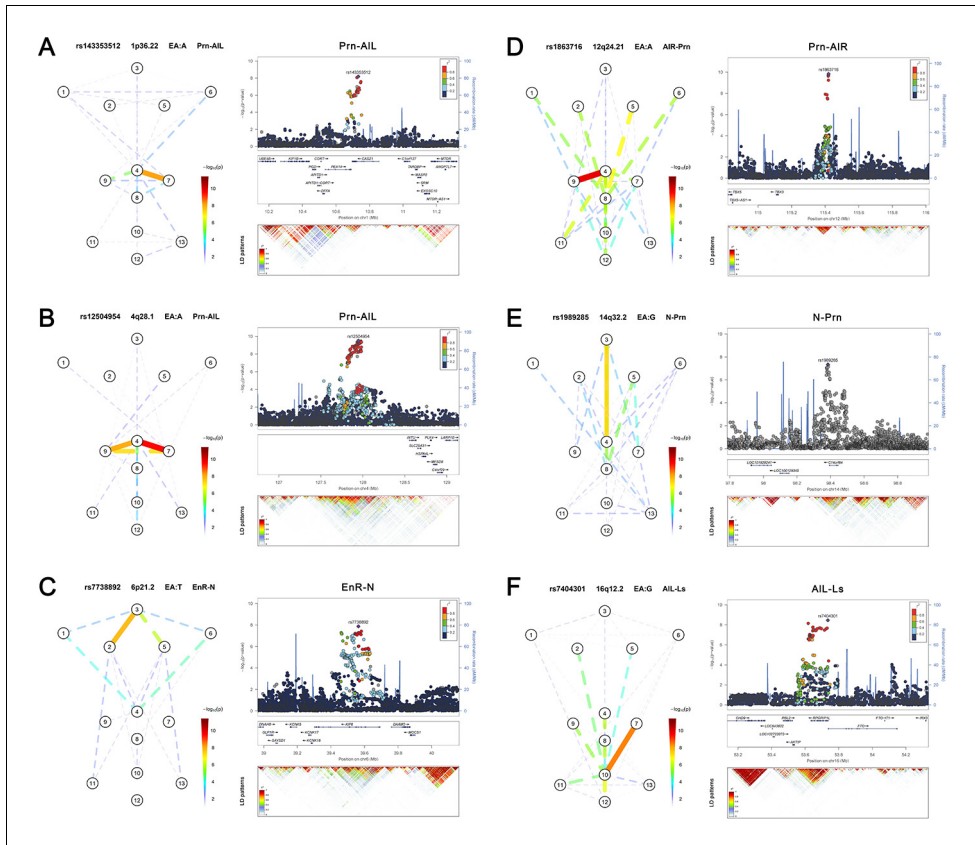

**Figure 2.** Six novel genetic loci associated with facial shape phenotypes and successfully replicated. The figure is composed by six parts, (**A**) 1p36.22 *CASZ1* region, (**B**) 4q28.1 *INTU* region, (**C**) 6p21.2 *KIF6* region, (**D**) 12q24.21 *TBX3* region, (**E**) 14q32.2 *C14orf64* region, and (**F**) 16q12.2 *RPGRIP1L* region. Each part includes two figures, Face map (left) denoting all of the top-SNP-associated phenotypes weighted by the statistical significance (-log$_{10}$P). The dashed line means that the P value was nominally significant (p<0.01) but did not reach study-wide suggestive significance (p>5e-8). LocusZoom (right) shows regional association plots for the top-associated facial phenotypes with candidate genes aligned below according to the chromosomal positions (GRCh37.p13) followed by linkage disequilibrium (LD) patterns (r$^2$) of EUR in the corresponding regions. Similar figures for all 24 study-wide suggestively significant loci are provided in *Figure 2—figure supplements 1–24*.

The online version of this article includes the following figure supplement(s) for figure 2:

**Figure supplement 1.** Overviews of association, selection, LD and regulation patterns in 24 associated regions.

**Figure supplement 2.** Association, selection, LD and regulation patterns for rs12504954 located around *INTU*.

**Figure supplement 3.** Association, selection, LD and regulation patterns for rs7738892 located around *KIF6*.

**Figure supplement 4.** Association, selection, LD and regulation patterns for rs1863716 located around *TBX3*.

**Figure supplement 5.** Association, selection, LD and regulation patterns for rs1989285 located around *C14orf64*.

**Figure supplement 6.** Association, selection, LD and regulation patterns for rs7404301 located around *RPGRIP1L*.

**Figure supplement 7.** Association, selection, LD and regulation patterns for rs200243292 located around *ARHGEF19*.

**Figure supplement 8.** Association, selection, LD and regulation patterns for rs77142479 located around *RPE65*.

**Figure supplement 9.** Association, selection, LD and regulation patterns for rs10202675 located around *LRRTM4*.

**Figure supplement 10.** Association, selection, LD and regulation patterns for rs2884836 located around *KIAA1715.*

**Figure supplement 11.** Association, selection, LD and regulation patterns for rs113663609 located around *CMSS1.*

**Figure supplement 12.** Association, selection, LD and regulation patterns for rs2225718 located around *RNF144B*.

**Figure supplement 13.** Association, selection, LD and regulation patterns for rs1700048 located around *CSMD1*.

**Figure supplement 14.** Association, selection, LD and regulation patterns for rs9642796 located around *DCAF4L2.*

**Figure supplement 15.** Association, selection, LD and regulation patterns for rs201719697 located around *SUPV3L1.*

*Figure 2 continued on next page*

nasion, subnasale, pronasale, left endocanthion and right endocanthion. The variance explained was potentially overestimated in RS cohort since the SNP selection and model building were conducted using the same dataset. However, potential overfitting is unexpected to have a drastic effect, given

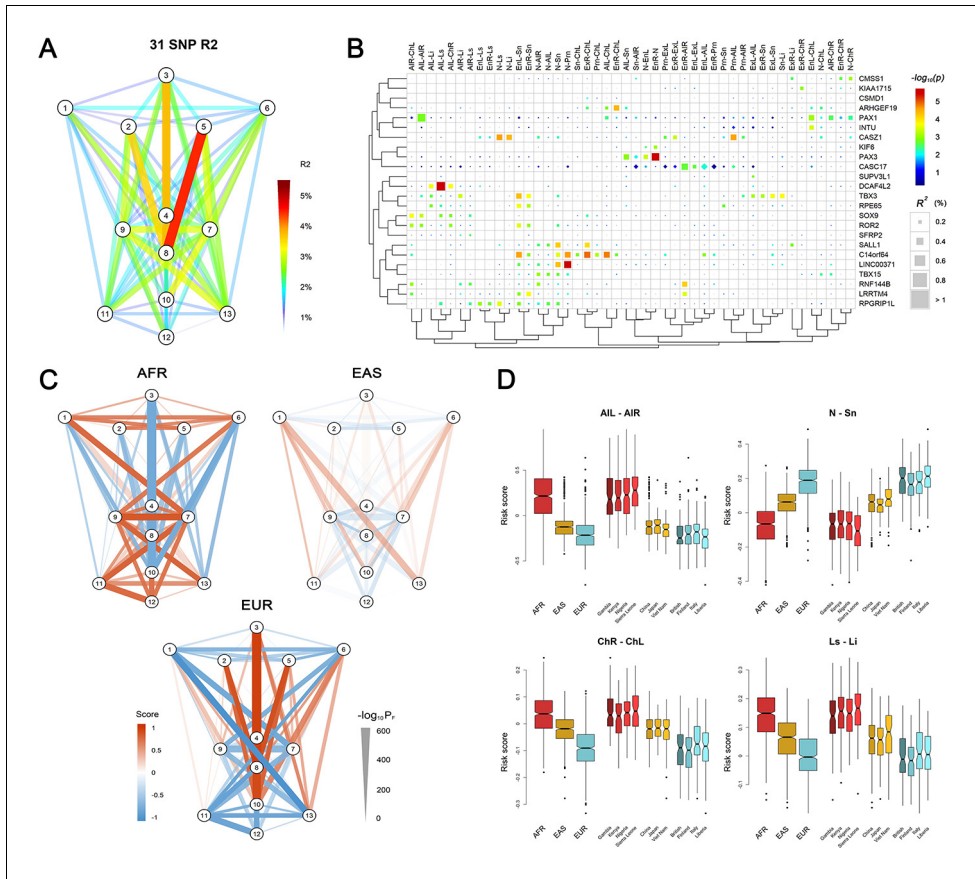

**Figure 3.** Polygenic scores of 78 facial phenotypes. (**A**) Facial phenotype variance is explained by a multivariable model consisting of 31 face-associated SNPs with independent effects. (**B**) Unsupervised clustering reveals the details on the explained phenotype variance, where the SNPs were attributed to the candidate genes in the associated regions. All phenotypes with the total explained variance > 2% are shown. SNPs were colored according to their raw p-values and those passing FDR < 0.05 were indicated in squares. (**C**) Mean standardized polygenic scores calculated by applying the multivariable model trained in the RS discovery sample to Sub-Saharan Africans (AFR), East Asians (EAS) and Europeans (EUR) in the 1000 Genomes Project data. (**D**) Examples of the distributions of the face scores in samples of different ancestry origin/country.

The online version of this article includes the following figure supplement(s) for figure 3:

**Figure supplement 1.** Effects of NSCL/P-associated SNPs to 78 facial phenotypes.

that some of the selected SNPs showed weaker effects in RS than in other cohorts used (*Supplementary file 1* - Table 7). We then used the fitted models to derive a set of polygenetic face scores for EUR, AFR and East Asian (EAS) subjects (total N = 1,668) from the 1000-Genomes Project. The polygenic face scores based on the 31 SNPs were largely reversely distributed between EUR and AFR, with EAS situated in between (*Figure 3C*). EUR showed the longest distances among the vertical phenotypes, while AFR had the widest of the horizontal phenotypes. EAS was consistently in-between with the most differential phenotype scores observed for nose wing breadth (AFR > EAS > EUR) and the nose length (EUR > EAS > AFR). An exception was for mouth phenotypes, where AFR had the largest values in both mouth width and lip thickness (*Figure 3D*). The nose-related score distribution, for example nose breadth and length, thus largely assembled the facial features in major continental groups, consistent with the hypotheses regarding adaptive evolution of human facial morphology (*Noback et al., 2011*; *Weiner, 1954*), for example nose width and nose length have been found to correlate with temperature and humidity. It should be noted, however, that the possibility of potential bias cannot be excluded due to the fact that the discovery GWAS was conducted in only Europeans. Notably, the polygenic scores of the NSCL/P-associated SNPs explained up to 1.2% of the age- and sex- adjusted phenotypic variance for phenotypes mostly between nose and mouth, for example right alare – right cheilion and pronasale – right cheilion (*Figure 3—figure supplement 1*).

## Gene ontology of face-associated genetic loci

A gene ontology enrichment analysis for the 24 face-associated genetic loci (*Table 1*) highlighted a total of 67 biological process terms and seven molecular function terms significantly enriched with different genes group (FDR < 0.01 or q value < 0.01, *Supplementary file 1* - Table 15), with the top significant terms being 'embryonic digit morphogenesis' and 'embryonic appendage morphogenesis' (FDR $< 1 \times 10^{-8}$, *Figure 4—figure supplement 1*). Biological process terms were categorized in four broader categories, 'morphogenesis', 'differentiation', 'development' and 'regulation'. These results underlined the important role of embryonic developmental and regulatory processes in shaping human facial morphology. Furthermore, *cis*-eQTL effects of the 494 face-associated SNPs at the 24 loci (*Supplementary file 1* - Table 7) were investigated using the GTEx data. Significant multi-tissue *cis*-eQTL effects (adjusted p<0.05) were found around three genes, *ARHGEF19*, *RPE65*, and *TBX15* (*Supplementary file 1* - Table 16). No significant tissue-specific eQTL effects were observed, likely due to the lack of cell types indexed within the GTEx database that are directly involved in craniofacial morphology.

## Expression of face-associated genetic loci in embryonic cranial neural crest cells

Embryonic cranial neural crest cells (CNCCs) arise during weeks 3–6 of human gestation from the dorsal part of the neural tube ectoderm and migrate into the branchial arches. They later form the embryonic face, consequently establishing the central plan of facial morphology and determining species-specific and individual facial variation (*Bronner and LeDouarin, 2012*; *Cordero et al., 2011*). Recently, CNCCs have been successfully derived in-vitro from human embryonic stem cells or induced pluripotent stem cells (*Prescott et al., 2015*). Taking advantage of the available CNCC RNA-seq data (*Prescott et al., 2015*) and other public RNA-seq datasets compiled from GTEx (*Lonsdale et al., 2013*) and ENCODE (*ENCODE Project Consortium, 2012*), we compared 24 genes, which were nearest to the top-associated SNPs at the 24 face-associated genetic loci identified in our discovery GWAS meta-analysis, with the background genes over the genome for their expression in CNCCs and 49 other cell types. In 89% of 9 embryonic stem cells, 65% of 20 different tissue cells, and 15% of 20 primary cells these 24 genes showed significantly higher expression in all cell types (p<0.05) after multiple testing correction compared to the background genes (*Figure 4A*), and the most significant difference was observed in CNCCs (p=3.8 $\times 10^{-6}$). Furthermore, these 24 genes also showed preferential expression in CNCCs compared to their expression in other cell types, that is in 65% of 20 different tissue cells, in 60% of 20 primary cells, and in 22% of 9 embryonic stem cells (*Figure 4A*). Specifically, 16 (66.7%) of these 24 genes showed preferential expression in CNCCs compared to 49 other cell types. The top-ranked three preferentially expressed genes were *PAX3* (p=1.3 $\times 10^{-29}$), *INTU* (p=8.3 $\times 10^{-25}$), and *ROR2* (p=4.5 $\times 10^{-23}$, *Figure 4B*).

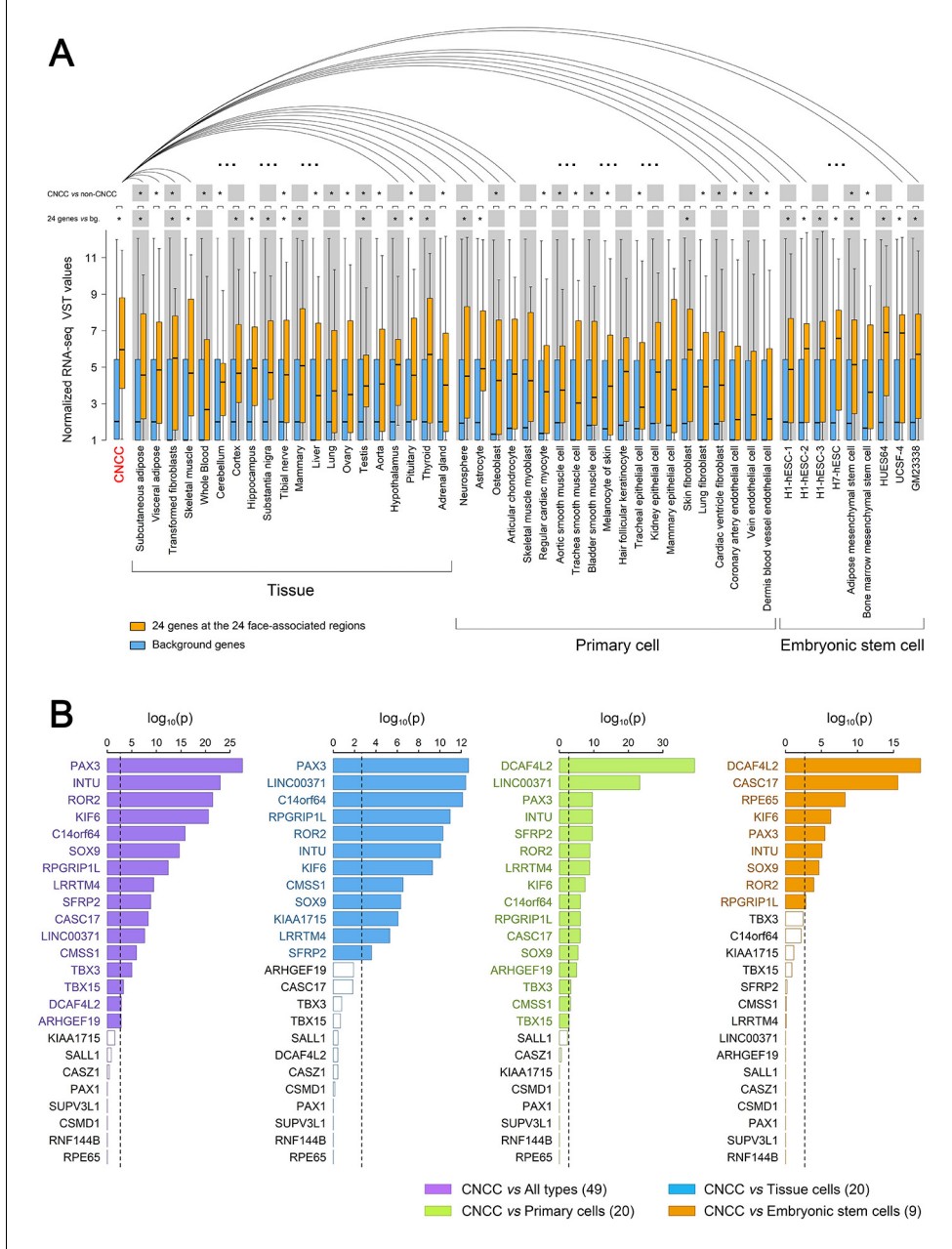

**Figure 4.** Expression of 24 genes located at the identified 24 face-associated genetic loci in 50 cell types. Expression levels of the genes nearest to the 24 top-associated SNPs (see *Table 1*) in 50 cell types were displayed in boxplots based on published data from the study of *Prescott et al. (2015)*, the GTEx database (*Lonsdale et al., 2013*) and the ENCODE database (*ENCODE Project Consortium, 2012*). The cell types included CNCC, 20 other tissue cells, 20 primary cells and nine embryonic stem cells. (**A**) Boxplots of normalized RNA-seq VST values for the 24 genes (in orange) and all genome background genes (~50,000 genes, in blue). Expression difference between the genome background genes and the 24 genes was tested in each cell type using the unpaired Wilcoxon rank-sum test. Distribution of the expression levels in CNCC showed smaller variation than those of several embryonic stem cells which showed higher median. The expression of the 24 genes in CNCCs was also iteratively compared with that in other cell types using paired Wilcoxon rank-sum test. Statistical significance was indicated: *$p<0.05$ after multiple correcting test of cell lines amount. (**B**) Difference significance of normalized RNA-seq VST values in each of 24 genes were displayed using barplots between CNCCs and all 49 types of cells (purple), CNCCs and 20 tissue cells (blue), CNCCs and 20 primary cells (green), CNCCs and nine embryonic stem cells (orange). The difference between the value of CNCCs and other cell type

*Figure 4 continued on next page*

*Figure 4 continued*

groups was tested using the one-sample Student's t test. Dotted line represents Bonferroni corrected significant threshold (p<0.0021). Significant gene labels were in color compared to non-significant gene labels in black. The online version of this article includes the following figure supplement(s) for figure 4:

**Figure supplement 1.** Gene ontology enrichment analysis for 24 face-associated genes.

Furthermore, some mid-preferentially expressed genes like *DCAF4L2* and non-preferentially expressed genes like *RPE65* became top-ranked preferentially expressed genes when testing sub-groups of cell types (*Figure 4B*). Although not necessarily is the nearest gene the causative gene and not necessarily is there only one causative gene per locus, the observation of significant preferential expression of the nearby genes considered in this analysis does support the hypothesis that variation of facial shape may indeed originate during early embryogenesis (*Claes et al., 2018*), and provide a priority list in different stages of cell differentiation with genes likely involved in embryo ectoderm derived phenotypes such as facial variation for future in-vivo functional studies.

## Cis-regulatory signals in face-associated genetic loci

It is becoming increasingly clear that c*is*-regulatory changes play a central role in determining human complex phenotypes (*Carroll, 2008*; *Wray, 2007*). In our study, apart from those at *KIF6* and *ROR2*, all identified face-associated DNA variants were non-exonic. With preferential expression pattern validated in CNCCs, we explored the potential *cis*-regulatory activity of the face-associated DNA variants using the CNCC epigenomic mapping datasets (*Prescott et al., 2015*). This includes ChIP-seq data on transcription factor (*TFAP2A* and *NR2F1*) binding, general coactivator (p300) binding, or histone modifications associated with active regulatory elements (H3K4me1, H3K4me3, and H3K27ac), as well as ATAC-seq data on chromatin accessibility. Abundant entries of *cis*-regulatory signals were detected surrounding the face-associated loci. In particular, 6 (25%) of the 24 face-associated loci, that is 1p36.13 *ARHGEF19*, 2q36.1 *PAX3*, *CMSS1*, 4q28.1 *INTU*, 16q12.1 *SALL1*, and 16q12.2 *RPGRIP1L*, showed significant enrichment of *cis*-regulatory entries at the top 0.1% of the genome level in at least one epigenomic tracks after Bonferroni correction (adjusted p<0.05, *Figure 5—figure supplement 1*). We selected 5 SNPs as candidate regulatory SNPs (*Figure 5A and B*, *Supplementary file 1* - Table 17) as they not only overlapped with the putative enhancers defined in CNCCs (*Prescott et al., 2015*) and penis foreskin melanocyte primary cells (*Chadwick, 2012*; *ENCODE Project Consortium, 2012*), but also showed high LD ($r^2 > 0.7$) with at least one of the face-associated SNPs at the study-wide suggestive significance level. These five candidate regulatory SNPs were further studied via functional experiments as described in the following section.

## Confirmation of cis-regulatory enhancer effects in neural crest progenitor cells

We performed in-vitro luciferase reporter assays to quantify the potential regulatory effect of the five face-associated SNPs suggested by the in-silico analyses outlined above as regulatory candidates that is rs13410020 and rs2855266 from the 2q36.1 *PAX3* region, rs17210317 and rs6828035 from the 4q28.1 *INTU* region, and rs4783818 from the 16q12.2 *RPGRIP1L* region (*Supplementary file 1* - Table 17). Luciferase reporter assays in CD271+ neural crest progenitor cells demonstrated that both SNPs in 2q36.1 and 4q28.1, respectively, showed statistically significant allele-specific effects on enhancer activity (*Figure 5C*). In the 4q28.1 region, the derived SNP alleles were associated with decreased luciferase expression (33.5 ×~ 10.4 × , *Supplementary file 1* - Table 18) as well as a reduced length of some nose features (−0.087 < beta < −0.078, *Supplementary file 1* - Table 7). In the 2q36.1 region, the derived SNP alleles were associated with increased luciferase expression (1.5 ×~ 3.7 × , *Supplementary file 1* - Table 18) as well as an increased length of some nasion-related features (0.131 < beta < 0.149, *Supplementary file 1* - Table 7). Using a luciferase assay for rs4783818 from the 16q12.2 region in CD271+ neural crest progenitor cells, we obtained an allele-specific effect on enhancer activity, although it was not statistically significant (*Figure 5C*). Additional luciferase reporter assays using G361 melanoma cells provided similar results, indicating that the regulatory activity associated with the DNA variants is conserved across the neural crest cell

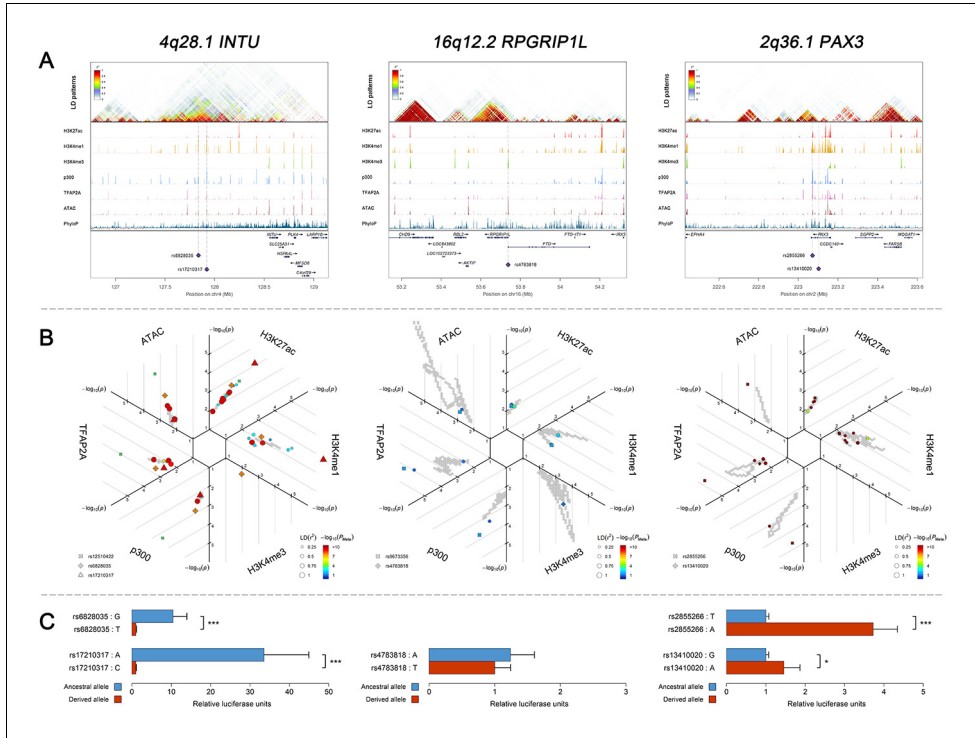

**Figure 5.** Three examples of genetic loci associated with facial shape phenotypes. Three examples of facial shape associated loci we discovered are depicted in details, including two novel loci (*INTU; RPGRIP1L*) and one (*PAX3*) that overlaps with previous face GWAS findings. The figure is composed by three layers (**A–C**) organized from the top to the bottom. (**A**) Denotes the linkage disequilibrium (LD) patterns ($r^2$) of EUR in the corresponding regions and epigenetic annotation of CNCC's regulatory elements in corresponding regions using WashU Epigenome Browser. Chip-seq profiles display histone modifications associated with active enhancers (H3K27ac, H3K4me1) or promoters (H3K4me3), and binding of general coactivator p300 and transcription factor TFAP2A. The ATAC-seq track shows chromatin accessibility. The PhyloP track indicates cross-species conservation. (**B**) SNPs in LD ($r^2 > 0.25$) with the top-associated SNP and all base-pairs in vicinity (within 20 kb, p<0.05) are displayed according to their log scaled quantile rank ($-\log_{10}p$, axis scale) in the circular figure, where the rank is calculated using the Chip-seq values in the whole genome. In addition, SNPs associated with facial phenotypes in our meta-analysis of GWASs were highlighted according to their association p values ($P_{Meta}$, color scale) and their LD ($r^2$) with the top-associated SNP in the region (points size). (**C**) Shows results of luciferase reporter assays in CD271+ neural crest progenitor cells by which we tested allele-specific enhancer activity of putative enhancers surrounding the selected SNPs using the Student's t-test. Data are presented as relative luciferase activity (firefly/renilla activity ratio) fold change compared to the construct containing less active allele. Similar figures for layers A and B are provided for all face associated loci in *Figure 2—figure supplements 1–24*.

The online version of this article includes the following figure supplement(s) for figure 5:

**Figure supplement 1.** Overview of regulation activity of 24 face-associated loci in CNCCs.

lineage (*Supplementary file 1* - Table 18). These results provide evidence that the tested face-associated SNPs impact the activity of enhancer elements and thus have regulatory function.

## Discussion

This study represents the largest genome scan to date examining facial shape phenotypes in humans quantified from 3D facial images. We identified 24 genetic loci showing study-wide suggestive association with normal-range facial shape phenotypes in Europeans, including 17 novel ones, of which 10 were successfully replicated in additional multi-ethnic population cohorts including six novel loci. Significant gene enrichment in morphogenesis related pathways and cis-regulation signals with preferential CNCC expression activities were observed at these genetic loci, underlining their involvement in facial morphology. Our findings were additionally supported by a strong integration with

previous genetic association and functional studies. Moreover, via in-vitro cell culture experiments, we functionally exemplified the regulatory activity of selected face-associated SNPs at two newly identified (*INTU* and *RPGRIP1L*) and one established (*PAX3*) face shape genes.

The six replicated novel face-associated loci included 1p36.22 *CASZ1* 4q28.1 *INTU* 6p21.2 *KIF6*, 12q24.21 *TBX3*, 14q32.2 *C14orf64*, and 16q12.2 *RPGRIP1L*. For 4q28.1 *INTU* and 16q12.2 *RPGRIP1L*, we demonstrated an impact of selected DNA variants on enhancer activity. *INTU*, also known as planar cell polarity (PCP) effector, has been reported to compromise proteolytic processing of *Gli3* by regulating Hh signal transduction (*Zeng et al., 2010*). Notably, SNPs in the *GLI3* region have been previously associated with human nose wing breadth (*Adhikari et al., 2016*), supporting a link between *INTU* and facial morphogenesis. In addition, rare frameshift mutations in *INTU* have been found to cause human Oral-Facial-Digital (OFD) syndromes as characterized by abnormalities of the face and oral cavity (*Bruel et al., 2017*). A previous study found that *Rpgrip1l* mutant mouse embryos displayed facial abnormalities in cleft upper lips and hypoplastic lower jaws (*Delous et al., 2007*). *FTO*, a proximate gene to *RPGRIP1L*, has also been found essential for normal bone growth and bone mineralization in mice (*Sachse et al., 2018*). *TBX3*, a member of the *T-box* gene family, causes Ulnar-mammary syndrome which is a rare disorder characterized by distinct chin and nose shape together with many other abnormal morphological changes (*Bamshad et al., 1997*; *Joss et al., 2011*). Intriguingly, *TBX3* and *TBX15* associated facial phenotypes showed an asymmetric pattern, particularly for nose phenotypes (*TBX3*) and eye-mouth distances (*TBX15*). These two genes are thus good candidates for further studying the genetic basis of facial asymmetry and related disorders. A synonymous variant (rs7738892) in the transcript coding region of *KIF6* was associated with facial traits. It has been shown that the homologous *Kif6* transcript sequence was necessary for spine development in zebrafish, likely due to its role in cilia, which is known to utilize neural crest cells for regulating ventral forebrain morphogenesis (*Buchan et al., 2014*). For the remaining two novel loci (*CASZ1* and *C14orf64*), there is current lack of existing literature evidence supporting their involvement in facial variation.

The four replicated previously reported face-associated loci included 2q36.1 *PAX3*, 4q31.3 *SFRP2*, 9q22.31 *ROR2*, and 20p11.22 *PAX1*. Their effects observed in the current study were all consistent with the previous findings, for example *PAX3* was associated with position of the nasion (*Liu et al., 2012*; *Paternoster et al., 2012*), *SFRP2* with distance between inner eye corners and alares (*Adhikari et al., 2016*; *Claes et al., 2018*), *PAX1* with nose wing breadth (*Adhikari et al., 2016*; *Shaffer et al., 2016*), and *ROR2* with nose size (*Pickrell et al., 2016*). The face-associated rs2230578 at 9q22.31 is in the 3'-UTR of *ROR2*. A *Ror2* knockout mouse has been used as a model for the developmental pathology of autosomal recessive Robinow syndrome, a human dwarfism syndrome characterized by limb shortening, vertebral and craniofacial malformations, and small external genitalia (*Schwabe et al., 2004*). Among the 11 non-replicated novel loci, two are worth mentioning: *KIAA1715* maps to about 100kbp upstream of the *HOXD* cluster (homeobox D cluster), which has been highlighted by two independent facial variation GWAS (*Claes et al., 2018*; *Pickrell et al., 2016*). *DCAF4L2* gene variants were previously associated with NSCL/P (*Leslie et al., 2017*; *Ludwig et al., 2012*; *Yu et al., 2017*) and we revealed a significant effect on the distance between alares and the upper-lip. We therefore regard it as likely that true positive findings exist among the 11 non-replicated novel loci. This is further supported by our finding that among the total of 14 non-replicated loci, three loci (21%) that is *TBX15* (*Claes et al., 2018*; *Pickrell et al., 2016*), *CASZ17* (*Claes et al., 2018*; *Pickrell et al., 2016*), and *SOX9* (*Cha et al., 2018*; *Claes et al., 2018*; *Pickrell et al., 2016*) correspond to previously identified face-associated loci. As such, their re-identification in our discovery GWAS meta-analysis represents a confirmation per se, even without significant evidence from our replication analysis. Moreover, pleiotropic effects of *TBX15* on skeleton, for example stature, were previously demonstrated by a large-sized GWAS (*Wood et al., 2014*). Loss of function of *TBX15* in mice has been shown to cause abnormal skeletal development (*Singh et al., 2005*). Thus, the fact that these three previously reported face-associated loci were identified in our discovery GWAS, but were not confirmed in our replication analysis, likely is explained by relatively low statistical power, especially for the European replication dataset (N = 1,101) being most relevant here as these three genes were highlighted in Europeans in the previous and in our study. Moreover, for the European QIMR samples in the replication set, no direct replication analysis was possible due to the availability of 2D images, but not 3D images as in the

discovery dataset. Therefore, future studies shall further investigate these 11 novel face-associated loci.

Of the previously identified face-associated genetic loci, several candidate genes for example *Pax3* (*Zalc et al., 2015*), *Prdm16* (*Bjork et al., 2010*), *Tp63* (*Thomason et al., 2008*), *Sfrp2* (*Satoh et al., 2008*), and *Gli3* (*Hui and Joyner, 1993*) have been shown to play important roles in cranial development of embryonic mice. Moreover, in adult mice only *Edar* has been investigated for its effect on normal facial variation (*Adhikari et al., 2016*), but the respective face-associated missense *EDARV370A* is nearly absent in Europeans (*Kamberov et al., 2013*). Here we showed via direct experimental evidence in neural crest cells that face-associated SNPs in *INTU*, *PAX3*, and *RPGRIP1L* are likely involved in enhancer functions and regulate gene activities. Future studies shall demonstrate functional evidence for more SNPs and genes identified here or previously to be significantly association with facial shape phenotypes. As demonstrated here, this shall go beyond evidence on the bioinformatics level (*Claes et al., 2018*) and shall involve direct functional testing of the associated DNA variants in suitable cell lines or mouse models.

Our results indirectly support the hypothesis that the facial differentiation observed among worldwide human populations is shaped by positive selection. For some facial features, it is reasonable to speculate a primary selective pressure on the morphological effects of the associated variants, although many of these variants may have pleiotropic effects on other fitness-related phenotypes. For example, nasal shape, which is essential for humidifying and warming the air before it reaches the sensitive lungs, has been found to significantly correlate with climatic variables such as temperature and humidity in human populations (*Noback et al., 2011*). Notably, many of the SNPs highlighted in our study are involved in nasal phenotypes. The overall effect of these SNPs as measured by polygenic scores largely assembled the nasal shape differences among major continental groups, which implies that human facial variation in part is caused by genetic components that are shared between major human populations. However, there are existing examples of population-specific effects, for example, *EDAR* clearly demonstrates an Asian specific effect, which impacts on various ectoderm-derived appearance traits such as hair morphology, size of the breast, facial shape in Asians (*Adhikari et al., 2015*; *Kamberov et al., 2013*; *Kimura et al., 2009*; *Tan et al., 2013*). The respective face-associated missense DNA variant is rare in European and African populations, demonstrating regional developments in Asia.

Nevertheless, the polygenic face scores only explained small proportions of the facial phenotype variance with up to 4.6%. This implies that many more face-predictive genetic information remains to be discovered in future studies, which - in case enough - may provide the prerequisite for practical applications of predicting human facial information from genomic data such as in forensics or anthropology. On the other hand, the increasing capacity of revealing personal information from genomic data may also have far-reaching ethical, societal and legal implications, which shall be broadly discussed by the various stakeholders alongside the genomic and technological progress made here and to be made in future studies.

In summary, this study identified multiple novel and confirmed several previously reported genetic loci influencing facial shape variation in humans. Our findings extend knowledge on natural selection having shaped the genetic differentiation underlying human facial variation. Moreover, we demonstrated the functional effects of several face-associated genetic loci as well as face-associated DNA variants with in-silico bioinformatics analyses and in-vitro cell line experiments. Overall, our study strongly enhances genetic knowledge on human facial variation and provides candidates for future in-vivo functional studies.

## Materials and methods

### Cohort details

The four discovery cohorts, totaling 10,115 individuals, were Rotterdam Study (RS, N = 3,193, North-Western Europeans from the Netherlands), TwinsUK (N = 1,020, North-Western Europeans from the UK), ALSPAC (N = 3,707, North-Western Europeans from the UK), and Pittsburgh 3D Facial Norms study (PITT, N = 2,195, European ancestry from the United States). The three replication cohorts, totaling 7917 individuals, were CANDELA Study (N = 5,958, Latin Americans with ancestry admixture estimated at 48% European, 46% Native American and 6% African), Xinjiang Uyghur

Study (UYG, N = 858, Uyghurs from China with ancestry admixture estimated at 50% East Asian and 50% European), and Queensland Institute of Medical Research Study (QIMR, N = 1,101, North-Western European ancestry from Australia). Three-dimensional facial surface images were acquired in all discovery cohorts and UYG, while frontal 2-dimensional photographs were used in CANDELA and QIMR.

## Rotterdam Study (RS)

The RS is a population based cohort study of 14,926 participants aged 45 years and older, living in the same suburb of Rotterdam, the Netherlands (*Hofman et al., 2013*). The present study includes 3193 participants of Dutch European ancestry, for whom high-resolution *3dMDface* digital photographs were taken. Genotyping was carried out using the Infinium II HumanHap 550K Genotyping BeadChip version 3 (Illumina, San Diego, California USA). Collection and purification of DNA have been described previously (*Kayser et al., 2008*). All SNPs were imputed using MACH software (www.sph.umich.edu/csg/abecasis/MaCH/) based on the 1000-Genomes Project reference population information (*Abecasis et al., 2012*). Genotype and individual quality controls have been described in detail previously (*Lango Allen et al., 2010*). After all quality controls, the current study included a total of 6,886,439 autosomal SNPs (MAF > 0.01, imputation R2 > 0.8, SNP call rate > 0.97, HWE > 1e-4). The Rotterdam Study has been approved by the Medical Ethics Committee of the Erasmus MC (registration number MEC 02.1015) and by the Dutch Ministry of Health, Welfare and Sport (Population Screening Act WBO, license number 1071272–159521 PG). The Rotterdam Study has been entered into the Netherlands National Trial Register (NTR; R; www.trial-register.nl) an) and into the WHO International Clinical Trials Registry Platform (ICTRP; P; www.who.int/ictrp/network/primary/en/) und under shared catalogue number NTR6831. All participants provided written informed consent to participate in the study and to have their information obtained from treating physicians.

## TwinsUK study

The TwinsUK study included 1020 phenotyped participants (all female and all of Caucasian ancestry) within the TwinsUK adult twin registry based at St. Thomas' Hospital in London. All participants provided fully informed consent under a protocol reviewed by the St. Thomas' Hospital Local Research Ethics Committee. Genotyping of the TwinsUK cohort was done with a combination of Illumina HumanHap300 and HumanHap610Q chips. Intensity data for each of the arrays were pooled separately and genotypes were called with the Illuminus32 calling algorithm, thresholding on a maximum posterior probability of 0.95 as previously described (*The MuTHER Consortium, 2011*). Imputation was performed using the IMPUTE 2.0 software package using haplotype information from the 1000 Genomes Project (Phase 1, integrated variant set across 1092 individuals, v2, March 2012). After all quality controls, the current study included a total of 4,699,858 autosomal SNPs (MAF > 0.01, imputation R2 > 0.8, SNP call rate > 0.97, HWE > 1e-4) and 1020 individuals.

## ALSPAC study

The Avon Longitudinal Study of Parents and Children (ALSPAC) is a longitudinal study that recruited pregnant women living in the former county of Avon in the United Kingdom, with expected delivery dates between 1st April 1991 and 31st December 1992. The initial number of enrolled pregnancies was 14,541 resulting in 14,062 live births and 13,988 children alive at the age of 1. When the oldest children were approximately 7 years of age, additional eligible cases who had failed to join the study originally were added to the study. Full details of study enrolment have been described in detail previously (*Boyd et al., 2013*; *Fraser et al., 2013*; *Golding et al., 2001*). A total of 9,912 ALSPAC children were genotyped using the Illumina HumanHap550 quad genome-wide SNP genotyping platform. Individuals were excluded from further analysis based on incorrect sex assignments; heterozygosity (0.345 for the Sanger data and 0.330 for the LabCorp data); individual missingness (>3%); cryptic relatedness (>10% IBD) and non-European ancestry (detected by a multidimensional scaling analysis seeded with HapMap two individuals). The resulting post-quality control dataset contained 8237 individuals. The post-quality control ALSPAC children were combined with the ALSPAC mothers cohort (*Fraser et al., 2013*) and imputed together using a subset of markers common to both the mothers and the children. The combined sample was pre-phased using ShapeIT (v2.r644)

(*Delaneau et al., 2012*) and imputed to the 1000 Genomes reference panel (Phase 1, Version3) (*Auton et al., 2015*) using IMPUTE3 V2.2.2 (*Howie et al., 2009*). The present study sample included 3707 individuals with 3D facial images, genotype and covariate data. Ethical approval for the study was obtained from the ALSPAC Ethics and Law Committee and the Local Research Ethics Committees. Please note that the study website contains details of all the data that is available through a fully searchable data dictionary and variable search tool: http://www.bristol.ac.uk/alspac/researchers/our-data/.

## Pittsburgh 3D Facial Norms (PITT) sample, United States

The current study included 2195 individuals from the 3D Facial Norms dataset (*Weinberg et al., 2016*). These individuals were recruited at four US sites (Pittsburgh, Seattle, Houston, and Iowa City), were of self-identified European ancestry, and ranged in age from 3 to 49 years. Exclusion criteria included any individuals with a history of significant facial trauma, congenital defects of the face, surgical alteration of the face, or any medical conditions affecting the facial posture. DNA was extracted from saliva samples and genotyped along with 72 HapMap control samples for 964,193 SNPs on the Illumina (San Diego, CA) HumanOmniEx- press+Exome v1.2 array plus 4,322 SNPs of custom content by the Center for Inherited Disease Research (CIDR). Samples were interrogated for genetic sex, chromosomal aberrations, relatedness, genotype call rate, and batch effects. SNPs were interrogated for call rate, discordance among duplicate samples, Mendelian errors among HapMap controls (parent-offspring trios), deviations from Hardy-Weinberg equilibrium, and sex differences in allele frequencies and heterozygosity. To assess population structure, we performed principal component analysis (PCA) using subsets of uncorrelated SNPs. Based on the scatterplots of the principal components (PCs) and scree plots of the eigenvalues, we determined that population structure was captured in four PCs of ancestry. Imputation of unobserved variants was performed using haplotypes from the 1000 Genomes Project Phase three as the reference. Imputation was performed using IMPUTE2. We used an info score of > 0.5 at the SNP level and a genotype probability of > 0.9 at the SNP-per-participant level as filters for imputed SNPs. Masked variant analysis, in which genotyped SNPs were imputed in order to assess imputation quality, indicated high accuracy of imputation. Institutional Review Board (IRB) approval was obtained at each recruitment center and all subjects gave written informed consent prior to participation: University of Pittsburgh IRB #PRO09060553 and IRB0405013; Seattle Children's IRB #12107; University of Iowa Human Subjects Office/IRB #200912764 and #200710721; UT Health Committee for the Protection of Human Subjects #HSC-DB-09–0508 and #HSC-MS-03–090.

## Xinjiang uyghur (UYG) study

The UYG samples were collected at Xinjiang Medical University in 2013–2014. In total, 858 individuals (including 333 males and 525 females, with an age range of 17–25) were enrolled. The research was conducted with the official approval from the Ethics Committee of the Shanghai Institutes for Biological Sciences, Shanghai, China. All participants had provided written consent. All samples were genotyped using the Illumina HumanOmniZhongHua-8 chips, which interrogates 894,517 SNPs. Individuals with more than 5% missing data, related individuals, and the ones that failed the X-chromosome sex concordance check or had ethnic information incompatible with their genetic information were excluded. SNPs with more than 2% missing data, with a minor allele frequency smaller than 1%, and the ones that failed the Hardy–Weinberg deviation test (p<1e-5) were also excluded. After applying these filters, we obtained a dataset of 709 samples with 810,648 SNPs for the Uyghurs. The chip genotype data were firstly phased using SHAPEIT (*O'Connell et al., 2014*). IMPUTE2 (*Howie et al., 2009*) was then used to impute genotypes at un-genotyped SNPs using the 1000 Genomes Phase three data as reference. Finally, a total of 6,414,304 imputed SNPs passed quality control and were combined with 810,648 genotyped SNPs for further analyses.

## CANDELA study

CANDELA samples (*Ruiz-Linares et al., 2014*) consist of 6630 volunteers recruited in five Latin American countries (Brazil, Colombia, Chile, México and Perú). Ethics approval was obtained from: The Universidad Nacional Autónoma de México (México), the Universidad de Antioquia (Colombia), the Universidad Peruana Cayetano Heredia (Perú), the Universidad de Tarapacá (Chile), the

Universidade Federal do Rio Grande do Sul (Brazil), and the University College London (UK). All participants provided written informed consent. Individuals with dysmorphologies, a history of facial surgery or trauma, or with BMI over 33 were excluded (due to the effect of obesity on facial features). DNA samples were genotyped on Illumina's Omni Express BeadChip. After applying quality control filters 669,462 SNPs and 6357 individuals were retained for further analyses (2922 males, 3435 females). Average admixture proportions for this sample were estimated as: 48% European, 46% Native American and 6% African, but with substantial inter-individual variation. After all genomic and phenotypic quality controls this study included 5958 individuals. The genetic PCs were obtained from the LD-pruned dataset of 93,328 SNPs using PLINK 1.9. These PCs were selected by inspecting the proportion of variance explained and checking scatter and scree plots. The final imputed dataset used in the GWAS analyses included genotypes for 9,143,600 SNPs using the 1000 Genomes Phase I reference panel. Association analysis on the imputed dataset were performed using the best-guess imputed genotypes in PLINK 1.9 using linear regression with an additive genetic model incorporating age, sex and five genetic PCs as covariates.

## Queensland Institute of Medical Research (QIMR) study

After phenotype and genotype quality control, data were available for 1101 participants who were mainly twins aged 16 years. All participants, and where appropriate their parent or guardian, gave informed consent, and all studies were approved by the QIMR Berghofer Human Research Ethics Committee. Participants were genotyped on the Illumina Human610-Quad and Core+Exome SNP chips. Genotype data were screened for genotyping quality (GenCall < 0.7), SNP and individual call rates (<0.95), HWE failure (p<1e-6) and MAF (<0.01). Data were checked for pedigree, sex and Mendelian errors and for non-European ancestry. Imputation was performed on the Michigan Imputation Server using the SHAPEIT/minimac Pipeline according to the standard protocols.

## Facial shape phenotyping

In our study, we focused on 13 facial landmarks available in all cohorts with 3-dimensional digital photographs. These included right (ExR) and left (ExL) exocanthion: points of bilateral outer canthus; right (EnR) and left (EnL) endocanthion: points of bilateral inner canthus; nasion (Nsn): the point where the bridge of the nose meets the forehead; pronasale (Prn): the point of the nose tip; subnasale (Sbn): the point where the base of the nasal septum meets the philtrum; right (AlR) and left (AlL) alare: points of bilateral nose wing; labliale superius (Ls): the point of labial superius; labiale inferius (Li): the point of labial inferius; right (ChR) and left (ChL) cheilion: points of bilateral angulus oris. Slightly different image processing algorithms were applied to 3dMD images to capture facial landmarks in the participating cohorts. In **RS** and **TwinsUK**, the raw 3D facial images were acquired using a 3D photographic scanning system manufactured by 3dMD (http://www.3dmd.com/). Participants were asked to keep their mouths closed and adopt a neutral expression during the acquisition of the 3D scans. Ensemble methods for automated 3D face landmarking were then used to derived 21 landmarks (*de Jong et al., 2018*; *de Jong et al., 2016*). In short, 3D surface model of participants' face obtained with commercial photogrammetry systems for faces called 3dMDface were input to the algorithm. Then a map projection of these face surfaces transformed the 3D data set into 3D relief maps and 2D images were generated from these 3D relief maps. These images were subjected to a 2D landmarking method like Elastic Bunch Graph Matching. Finally, registered 2D landmarks are mapped back into 3D, inverting the projection. 13 shared facial landmarks coordinate data were processed using GPA to remove variations due to scaling, shifting and rotation. Euclidean distances were calculated between all landmarks. Outliers (>3 sd) were removed before the subsequent analysis. Shapiro-Wilk test (*Royston, 1995*) was used to test the normality of phenotype distribution. At 15 years of age, the **ALSPAC** children were recalled (5253 attended) and 3D facial scans where taken using two high-resolution Konica Minolta VI-900 laser scanners (lenses with a focal length 14.5 mm). The set of left and right facial scans of each individual were processed, registered, and merged using a locally developed algorithm implemented as a macro in Rapidform 2006 (*Kau et al., 2004*; *Paternoster et al., 2012*; *Toma et al., 2009*). 22 landmarks were recorded manually. GPA was used to remove affine variations due to shifting, rotation and scaling. 4747 individuals had usable images (506 individuals did not complete the assessment or facial scans were removed due to poor quality). The coordinates of 22 facial landmarks were derived from the scans, of which 13 were shared across

all other studies. 3D Euclidean distances were calculated between the facial landmarks. 3D facial models from **PITT** were obtained with 3dMD digital stereo-photogrammetry systems. Facial landmarks were collected on the facial surfaces by trained raters and assessed for common placement errors during subsequent quality control. A set of 78 linear distance variables was then calculated from a subset of 13 of these landmarks, chosen because they were shared among all of the discovery cohorts in this study. These distances were screened for outliers. In **UYG**, the 3DMDface system ([www.3dmd.com/3dMDface](www.3dmd.com/3dMDface)) was used to collect high-resolution 3D facial images from participants. A dense registration was applied to align all the 3D images as described elsewhere (*Guo et al., 2013*). In brief, 15 salient facial landmarks were first automatically annotated based on the PCA projection of texture and shape information. Afterwards, a facial image of high quality and smooth shape surface was chosen as the reference, and its mesh was re-sampled to achieve an even density of one vertex in each 1mm $\times$ 1 mm grid. The reference face was then warped to register every sample face by matching all the 15 landmarks, via a non-rigid thin-plate spline (TPS) transformation. The mesh points of the reference face were then projected to the sample surface to find their one-to-one correspondents. The resulting points of projection were used to define the mesh of the sample facial surface. After the alignment, each sample face was represented by a set of 32,251 3D points and 15 target points were ready to be analyzed. A set of 78 linear distance variables was calculated from a subset of 13 of these 15 landmarks. In **CANDELA**, ordinal phenotyping was described in prior GWAS (*Adhikari et al., 2016*). In short, right side and frontal 2D photographs of participants were used to score 14 facial traits, Intra-class correlation coefficients (ICCs) were calculated to indicate a moderate-to-high intra-rater reliability of the trait scores. All photographs were scored by the same rater. In **QIMR**, two independent researchers manually evaluated all 2D portrait photos and located the coordinates for 36 facial landmarks. Satisfactory inter-rater concordance was obtained (Pearson's correlation coefficient, mean r = 0.76). The average landmark coordinates between the two researchers were used for Euclidian distance calculation.

## Phenotype characteristics

Phenotype characteristics were explored for gender and aging effect, phenotypic and genetic correlations, clustering, and twins-heritability. Gender and aging effects were estimated using beta, p, and $R^2$ values from linear models. Phenotypic correlations were estimated using Pearson's correlation coefficients. Genetic correlations were estimated using genome-wide SNPs and the multivariate linear mixed model algorithm implemented in the GEMMA software package (*Zhou & Stephens, 2014*). Manual clustering was according to the involvement of intra- and inter- organ landmarks. Unsupervised hierarchical clustering was conducted using the 1-abs(correlation) as the dissimilarity matrix and each iteration was updated using the Lance - Williams formula (*Ward, 1963*). we calculated twins heritability for TwinsUK and QIMR twins using the phenotypic correlation derived from monozygotic twins (MZ) and dizygotic twins (DZ), that is $h^2 = 2[cor(MZ) - cor(DZ)]$. SNP-based heritability was estimated for twins cohorts using the restricted estimated maximum likelihood method implemented in the GCTA software package (*Yang et al., 2011*).

## GWAS, meta-analysis, and replication studies

GWASs for face phenotypes were independently carried out in RS, TwinsUK, ALSPAC, PITT, CANDELA, UYG, and QIMR based on linear or logistic models assuming an additive allele effect adjusted for potential confounders in each cohorts, such as family relationships, sex, age, BMI and genetic PCs using software packages including PLINK (*Purcell et al., 2007*), GEMMA (*Zhou and Stephens, 2012*), and Rare Metal Worker (*Feng et al., 2014*). It should be noted that we used the scaled General Procrustes Analysis (GPA) to eliminate potential confounding effect of BMI although in some cohorts BMI was additionally included as a covariate, which shouldn't have affected the results of our meta-analysis. For **RS**, association tests were run between the 78 3D facial distance measurements (after scaled GPA) and 6,886,439 genotyped and imputed SNPs with MAF > 0.01, imputation R2 > 0.8, SNP call rate > 0.97 and HWE > 0.0001. GWAS was performed using linear regression under the additive genetic model while adjusting for sex, age, and the first four genomic PCs derived using GCTA (*Yang et al., 2011*). For TwinsUK, quality control was performed on SNPs (MAF > 0.01, imputation R2 > 0.8, SNP call rate > 0.99). One of each pair of monozygotic twins (MZ) were excluded from association tests to reduce relatedness. The final sample included 1020

individuals and 4,699,859 autosomal genotyped and imputed variants. GWAS was performed between the 78 3D facial distance measurements (after scaled GPA) and genotyped and imputed SNPs using linear regression under the additive genetic model while adjusting for age and the first four genomic PCs. For **ALSPAC**, 3941 study participants had both genotype and 3D facial phenotype data. GCTA was used to remove individuals with excessive relatedness (GRM > 0.05) and quality control was performed on SNPs (MAF > 0.01, imputation R2 > 0.8, SNP call rate > 0.99). The final sample included 3707 individuals and 4,627,882 autosomal genotyped and imputed SNPs. GWAS was performed between the 78 3D facial distance measurements (after scaled GPA) and genotyped and imputed SNPs using linear regression under the additive genetic model while adjusting for age, sex, and the first four genomic PCs using PLINK 1.9 (*Purcell et al., 2007*). For **PITT**, association tests were run between the 78 3D facial distance measurements (after scaled GPA) and 9,478,231 genotyped and imputed SNPs with MAF > 1% and HWE > 0.0001. GWAS was performed using linear regression under the additive genetic model while adjusting for sex, age, age$^2$, height, weight and the first four genomic PCs using PLINK 1.9. For the analysis of the X-chromosome, genotypes were coded 0, 1, and two as per the additive genetic model for females, and coded 0, two for males in order to maintain the same scale between sexes. Note that in the meta-analysis of all cohorts, the X-chromosome was not included. For **CANDELA**, PLINK 1.9 was used to perform the primary genome-wide association tests for each phenotype using multiple linear regression with an additive genetic model incorporating age, sex, BMI and five genetic PCs as covariates. Association analyses were performed on the imputed dataset with two approaches: using the best-guess imputed genotypes in PLINK, and using the IMPUTE2 genotype probabilities in SNPTEST v2.5. Both were consistent with each other and with the results from the chip genotype data. GWAS results with using best-guess genotypes were used in the subsequent meta-analysis. For **UYG**, GWAS were carried out using PLINK 1.9 and a linear regression model with additive SNP effects was used. Details was described in prior GWAS (*Qiao et al., 2018*). For **QIMR**, the GWAS was conducted using Rare Metal Worker with linear regression adjusted for sex, age, BMI and the first five genomic PCs.

Inverse variance fixed-effect meta-analysis were carried out using PLINK to combine GWAS results for samples of European origin and having 3dMDface data, that is RS, TwinsUK, ALSPAC, and PITT, which included 7,029,494 autosomal SNPs overlapping between the 4 discovery cohorts with physical positions aligned according to NCBI human genome sequence build GRCh37.p13. X chromosome was not included in meta-analyses. Summary statistics of all SNPs and all facial phenotypes from the GWAS meta-analysis from the four cohorts of European descent (RS, TwinsUK, ALSPAC and PITT) are available via figshare https://doi.org/10.6084/m9.figshare.10298396 and will be made available through the NHGRI-EBI GWAS Catalog https://www.ebi.ac.uk/gwas/downloads/summary-statistics. Meta-analysis results were visualized using superposed Manhattan plots and Q-Q plots. All genomic inflation factors were close to 1 and not further considered. Because many of the 78 facial phenotypes were correlated, we applied the matrix decomposition method described by *Li and Ji (2005)* to determine the effective number of independent phenotypes as 43. We then set our study-wide significant threshold at p<1.2×10$^{-9}$ using Bonferroni correction of 43 phenotypes, considering significant threshold for single phenotype GWAS as P<5×10$^{-8}$). We considered the threshold of 5×10$^{-8}$ as the threshold for replication studies in CANDELA, UYG, and QIMR. Since the 3dMDface data were available in UYG, exact replications were conducted, that is the genetic association was tested for the corresponding phenotypes in the meta-analysis. Since only 2D photos were available in CANDELA and QIMR, trait-wise replications were conducted to test H0: no association for all face phenotypes vs. H1: associated with at least one face phenotypes. We used a combined test of dependent tests to combine genetic association tests involving highly correlated face phenotypes (*Kost and McDermott, 2002*). In short, the test statistic of the combined test follows a scaled chi-squared distribution when tests are dependent. The mean of the $\chi^2$ values is:

$E(\chi^2) = 2k$, and the variance can be estimated as

$var(\chi^2) = 4k + 2\sum_{1 \leq i < j \leq k} cov(-2logP_i, -2logP_j)$, where

$k$ is the number of phenotypes, $P_i$ is the $p$ value of the $i$th test. The covariance term $cov(-2logP_i, -2logP_j)$ can be approximated by the following formula using the phenotype correlation matrix:

$$cov(-2logP_i, -2logP_j) \approx 3.263r_{ij} + 0.71r_{ij}^2 + 0.027r_{ij}^3$$

This approximation has been shown nearly identical to the exact value with difference less than 0.0002 for $-0.98 < r_{ij} < 0.98$ (*Kost and McDermott, 2002*). We then again used the combined test of dependent tests to combine replication evidence from all three cohorts. Multiple testing was corrected for the number of selected SNPs using the Bonferroni method. Regional linkage disequilibrium $r^2$ values were calculated in PLINK and visualized using R scripting. Regional association plots were produced using LocusZoom (*Pruim et al., 2010*).

## Genetic diversity and positive selection signal analysis

To test for signals of allele frequency diversity and positive selection in the face associated genomic regions, we derived two commonly used statistics in population genetics, that is the fixation index (FST) and the integrated haplotype score (iHS) for all SNPs in the associated regions. The FST is a parameter in population genetics for quantifying the differentiation due to genetic structure between a pair of groups by measuring the proportion of genetic diversity due to allele frequency differences among populations potentially as a result of divergent natural selection (*Holsinger and Weir, 2009*). We performed a genome-wide FST analysis for three pairs of populations (AFR-EAS, EAS-EUR, and EUR-AFR) in the samples from the 1000 Genome Project using the PLINK software. The empirical significance cutoff for each of the three population pairs was set to the value corresponding to the top 1% FST values in the genome. The iHS is the integrate of the extended haplotype homozygosity (EHH) statistic to highlight recent positive selected variants that have not yet reached fixation, and large values suggest the adaption to the selective pressures in the most recent stage of evolution (*Sabeti et al., 2002*; *Voight et al., 2006*). The EHH statistic was calculated for all SNPs in AFR, EAS and EUR samples. Then the iHS statistic, integral of the decay of EHH away from each SNP until EHH reaches 0.05, was calculated for all SNPs, in which an allele frequency bin of 0.05 was used to standardize iHS scores against other SNPs of the same frequency class. Finally, we calculated empirical significance threshold over the genome assuming a Gaussian distribution of iHS scores under the neutral model. The cutoff was based on the top 1% absolute iHS scores calculated in three samples respectively as the absolute score indicate the selection intensity.

## Multivariable and polygenic score (PS) analysis

Multivariable and polygenic score analyses were conducted in the RS cohort. For all SNPs showing study-wide suggestive significant association with face phenotypes ($p<5x10^{-8}$), we performed a multivariable regression analysis aiming to select SNPs that still be significant after regressing out the effects of the 24 lead SNPs by including them as covariates in the model, which resulted in 31 SNPs for the subsequent modeling. A forward step-wise regression was conducted to access the independent effects of the selected SNPs on sex- and age- adjusted face phenotypes, where the $R^2$ contribution of each SNP was estimated by comparing the current model fitting $R^2$ with the $R^2$ in a previous iteration. Next, we calculated PS for AFR, EUR, EAS individuals of the 1000 Genomes Project using the selected SNPs and the effect sizes estimated from the multivariable analysis, that is $PS = \sum_{i=1}^{n} \beta_i A_i$. The mean values and distributions of the standardized PS values for all face phenotypes in each continental group was visualized using a face map and multiple boxplots, in which the differences between groups were highlighted according to the p-values from the ANOVA test, which tests H0: PS values are the same in all continental groups and H1: PS values are different in at least one of the groups.

## Gene ontology analysis, expression analysis and SNP functional annotation

To further study the involved functions and regulation relationships of the genome-wide significantly associated SNP markers and candidate genes nearby, the genes located within 100 kb up- and down-stream of the SNPs were selected to perform biological process enrichment analysis based on the Gene Ontology (GO) database. The enrichment analysis was performed using the R package 'clusterProfiler' (*Yu et al., 2012*). RNA-seq data from the study of *Prescott et al. (2015)*, GTEx (*Lonsdale et al., 2013*) and ENCODE (*ENCODE Project Consortium, 2012*) were used to compare the differences in gene expression levels between our face-associated genes and all genes over the genome in CNCCs. We examined entries in GWAS catalog (*Welter et al., 2014*) to explore the potential pleiotropic effects in our face associated regions. The GTEx (*Lonsdale et al., 2013*) dataset

was used to annotate our face associated SNPs for significant single-tissue *cis*-eQTL effects in all 48 available tissues. We used the CNCC data from Prescott et al. and UCSC (*Aken et al., 2016*) to perform a variety of functional annotations for the face associated SNPs in non-coding regions using the WashU Epigenome Browser, including histone modifications (H3K27ac, H3K4me1 and H3K4me3 tracks), chromatin modifier (p300 tracks), transcription factors (TFAP2A tracks), chromatin accessibility (ATACseq tracks) and conserved regions (PhyloP (*Siepel et al., 2005*) tracks) from UCSC. To investigate the pattern of regulation within each association region in details, we calculated the log scaled quantile rank and variation of all LD SNP ($r^2 > 0.25$) in three replicated measurements from 6 types of Chip-seq data, and for all significant base-pair in vicinity (within 20 kb) of all top-associated SNPs. All analyses were conducted using R Environment for Statistical Computing (version 3.3.1) unless otherwise specified.

## Cell culture

Neural progenitors were derived from human induced pluripotent stem cells as described previously (male, age 57 years, derived from primary skin fibroblasts) (*Gunhanlar et al., 2018*). Written informed consent was obtained in accordance with the Medical Ethical Committee of the Erasmus University Medical Center. Neural progenitors contain a subpopulation of neural crest progenitor cells that were enriched by fluorescence-activated cell sorting for CD271 (BD Bioscience 560927, BD FACSAria III) (*Dupin and Coelho-Aguiar, 2013*; *Lee et al., 2007*). CD271 (p75) is a surface antigen marking neural crest progenitor cells which give rise to craniofacial tissues as well as melanocyte primary cells (*Morrison et al., 1999*). CD271+ cells were grown in DMEM:F12, 1% N2 supplement, 2% B27 supplement, 1% P/S, 20 ng/ml FGF at 37°C/5% $CO_2$. G361 cells were cultured in DMEM/10% FCS at 37°C/5% $CO_2$. The cell line was tested negative for mycoplasma contamination and its identity was authenticated by STR profiling using PowerPlex 16 System (Promega).

## In vitro functional experiments: Luciferase reporter assay

We explored the effect of the five selected SNPs on gene expression in CD271+ and G361 melanoma cells by a luciferase reporter assay. Custom 500 bp gBlocks Gene fragments (IDT) containing putative enhancers surrounding each of the five selected SNPs were cloned into a modified pGL3 firefly luciferase reporter vector (SV40 promoter and 3′UTR replaced by HSP promoter and 3′UTR). For each SNP, constructs containing both ancestral and derived alleles were tested. Equimolar amount of constructs (measured by qPCR detecting the firefly luciferase gene using GoTaq qPCR Master Mix (Promega) were co-transfected with renilla luciferase control vector into CD271+ and G361 cells using Lipofectamine LTX (Invitrogen). Luciferase activity was measured 48 hr after transfection using the Dual-Glo Luciferase Assay System (Promega). Data are presented as relative luciferase activity (firefly/renilla activity ratio) fold change compared to the construct containing the less active allele and represent at least three independent replicates. Student's t-test was performed to test differences in gene expression. The Key Resources Table summarizing reagents and resources used in the in-vitro functional experiments is included in the *Supplementary file 1* - Table 19 (see *Supplementary file 1*).

## Acknowledgements

We are grateful to all participants and their families who took part in the RS, TwinsUK, ALSPAC, PITT, UYG, CANDELA, and QIMR studies as well as the many individuals involved in the running of the studies and recruitment, which include midwives, interviewers, computer and laboratory technicians, clerical workers, research scientists, volunteers, managers, receptionists, nurses and other professional. MK additionally thanks Maren von Köckritz-Blickwede for earlier discussions on PAD knockout mice. QIMR would like to thank Natalie Garden and Jessica Adsett for data collection and Scott Gordon for data management.

The **ALSPAC study** was supported by The UK Medical Research Council and Wellcome (Grant ref: 102215/2/13/2) and the University of Bristol. A comprehensive list of grants funding for ALSPAC is available online (http://www.bristol.ac.uk/alspac/external/documents/grant-acknowledgements.pdf). This publication is the work of the authors and L.H. and E.S. will serve as guarantors for the contents of this paper. GWAS data was generated by Sample Logistics and Genotyping Facilities at Wellcome Sanger Institute and LabCorp (Laboratory Corporation of America) using support from

23andMe. The **TwinsUK study** was supported by the Wellcome Trust, Medical Research Council, European Union (FP7/2007-2013), the National Institute for Health Research (NIHR)-funded BioResource, Clinical Research Facility and Biomedical Research Centre based at Guy's and St Thomas' NHS Foundation Trust in partnership with King's College London. The **Rotterdam Study** is supported by the Erasmus MC and Erasmus University Rotterdam; the Netherlands Organization for Scientific Research (NWO); the Netherlands Organization for Health Research and Development (ZonMw); the Research Institute for Diseases in the Elderly (RIDE) the Netherlands Genomics Initiative (NGI); the Ministry of Education, Culture and Science; the Ministry of Health Welfare and Sport; the European Commission (DG XII); and the Municipality of Rotterdam. The generation and management of GWAS genotype data for the Rotterdam Study were executed by the Human Genotyping Facility of the Genetic Laboratory of the Department of Internal Medicine, Erasmus MC. For the **CANDELA study**, the following institutions kindly provided facilities for the assessment of volunteers: Escuela Nacional de Antropología e Historia and Universidad Nacional Autónoma de México (México); Pontificia Universidad Católica del Perú, Universidad de Lima and Universidad Nacional Mayor de San Marcos (Perú); Universidade Federal do Rio Grande do Sul (Brazil); 13° Companhia de Comunicações Mecanizada do Exército Brasileiro (Brazil). The **PITT study** received additional support from The Centers for Disease Control and Prevention (https://www.cdc.gov/) provided funding through the following grant: R01- DD000295. Funding for initial genomic data cleaning by the University of Washington was provided by contract #HHSN268201200008I from the National Institute for Dental and Craniofacial Research (http://www.nidcr.nih.gov/) awarded to the Center for Inherited Disease Research (CIDR). The **QIMR study** received additional support by an Australian Research Council Linkage Grant (LP 110100121 to Dennis McNevin, University of Canberra).

## Additional information

### Funding

| Funder | Grant reference number | Author |
|---|---|---|
| Horizon 2020 - Research and Innovation Framework Programme | 740580 (VISAGE) | Manfred Kayser |
| National Science Foundation of China | 91651507 | Fan Liu |
| Chinese Academy of Sciences | Strategic Priority Research Program (XDC010400100) | Fan Liu |
| China Scholarship Council | PhD Fellowship | Ziyi Xiong |
| Netherlands Organisation for Scientific Research | 1750102005011 | M Arfan Ikram |
| Wellcome Trust | | Timothy D Spector |
| Medical Research Council | 102215/2/13/2 | Evie Stergiakouli |
| National Institute of Dental and Craniofacial Research | U01-DE020078 | Mary L Marazita Seth M Weinberg |
| Leverhulme Trust | F/07 134/DF | Andres Ruiz-Linares |
| National Natural Science Foundation of China | 91631307 | Sijia Wang |
| National Natural Science Foundation of China | 91731303 | Shu-Hua Xu |
| National Natural Science Foundation of China | 30890034 | Li Jin |
| National Health and Medical Research Council | APP103119 | Nicholas G Martin |
| National Health and Medical Research Council | Fellowship (APP1103623) | Sarah E Medland |

| | | |
|---|---|---|
| National Natural Science Foundation of China | 31771388 | Shu-Hua Xu |
| National Natural Science Foundation of China | 315014 | Shu-Hua Xu |
| National Natural Science Foundation of China | 31711530221 | Shu-Hua Xu |
| National Natural Science Foundation of China | 31271338 | Li Jin |
| National Institute of Dental and Craniofacial Research | R01-DE027023 | John R Shaffer<br>Seth M Weinberg |
| National Institute of Dental and Craniofacial Research | R01-DE016148 | Mary L Marazita<br>Seth M Weinberg |
| National Institute of Dental and Craniofacial Research | X01-HG007821 | Eleanor Feingold<br>Mary L Marazita<br>Seth M Weinberg |
| Netherlands Organisation for Scientific Research | 911-03-012 | M Arfan Ikram |
| Ministry of Science and Technology of the People's Republic of China | National Key R&D Program of China (2017YFC083501) | Fan Liu |
| Shanghai Municipal Science and Technology Commission | Major Project 2017SHZDZX01 | Sijia Wang<br>Shu-Hua Xu |
| BBSRC | BB/I021213/1 | Andres Ruiz-Linares |
| Universidad de Antioquia | CPT-1602, Estrategia para sostenibilidad 2016-2018 grupo GENMOL | Gabriel Bedoya |
| Science and Technology Commission of Shanghai Municipality | 16JC1400504 | Li Jin<br>Sijia Wang |
| Chinese Academy of Sciences | QYZDJ-SSW-SYS009 | Shu-Hua Xu |
| National Basic Research Program of China | 2015FY111700 | Li Jin<br>Sijia Wang |
| National Key Research and Development Program China | 2016YFC0906403 | Shu-Hua Xu |
| National Key Research and Development Program China | 2017YFC0803501-Z04 | Kun Tang |
| National Program for Top-notch Young Innovative Talents of The "Wanren Jihua" Project China | | Shu-Hua Xu |
| Thousand Talents Plan | | Sijia Wang |
| Max Planck-CAS Paul Gerson Unna Independent Research Group Leadership Award | | Sijia Wang |
| Medical Research Council | MC_UU_00011/1 | Laurence J Howe<br>Sarah Lewis<br>Gemma C Sharp<br>Evie Stergiakouli |
| Medical Research Council | | Timothy D Spector |
| Netherlands Organisation for Scientific Research | 175.010.2005.011 | André G Uitterlinden |
| Netherlands Organisation for Scientific Research | 911-03-012 | André G Uitterlinden |
| Netherlands Organisation for Scientific Research | 050-060-810 | André G Uitterlinden |

| Netherlands Genomics Initiative | 050-060-810 | André G Uitterlinden |
|---|---|---|

The funders had no role in study design, data collection and interpretation, or the decision to submit the work for publication.

## Author contributions

Ziyi Xiong, Data curation, Software, Formal analysis, Funding acquisition, Investigation, Visualization, Methodology, Manuscript preparation and revision, Final approval of the version to be published; Gabriela Dankova, Data curation, Formal analysis, Investigation, Visualization, Methodology, Data acquisition, Manuscript preparation and revision, Final approval of the version to be published; Laurence J Howe, Data curation, Formal analysis, Investigation, Funding acquisition, Manuscript preparation, Final approval of the version to be published; Myoung Keun Lee, Pirro G Hysi, Data curation, Formal analysis, Investigation, Final approval of the version to be published; Markus A de Jong, Data curation, Software, Formal analysis, Investigation, Data acquisition, Final approval of the version to be published; Gu Zhu, Kaustubh Adhikari, Data curation, Formal analysis, Validation, Investigation, Final approval of the version to be published; Dan Li, Formal analysis, Validation, Investigation, Final approval of the version to be published; Yi Li, Bo Pan, Formal analysis, Investigation, Final approval of the version to be published; Eleanor Feingold, Data curation, Formal analysis, Final approval of the version to be published; Mary L Marazita, Resources, Data curation, Data acquisition, Final approval of the version to be published; John R Shaffer, Resources, Data curation, Formal analysis, Funding acquisition, Final approval of the version to be published; Kerrie McAloney, Resources, Data curation, Formal analysis, Validation, Final approval of the version to be published; Shu-Hua Xu, Li Jin, Resources, Supervision, Funding acquisition, Validation, Project administration, Data acquisition, Final approval of the version to be published; Sijia Wang, Resources, Data curation, Supervision, Funding acquisition, Validation, Project administration, Data acquisition, Final approval of the version to be published; Femke MS de Vrij, Bas Lendemeijer, Resources, Investigation, Final approval of the version to be published; Stephen Richmond, Resources, Supervision, Investigation, Data acquisition, Manuscript preparation, Final approval of the version to be published; Alexei Zhurov, Data curation, Formal analysis, Investigation, Data acquisition, Final approval of the version to be published; Sarah Lewis, Resources, Data curation, Investigation, Funding acquisition, Final approval of the version to be published; Gemma C Sharp, Resources, Data curation, Investigation, Funding acquisition, Project administration, Final approval of the version to be published; Lavinia Paternoster, Resources, Data curation, Supervision, Project administration, Final approval of the version to be published; Holly Thompson, Resources, Data curation, Supervision, Final approval of the version to be published; Rolando Gonzalez-Jose, Maria Catira Bortolini, Samuel Canizales-Quinteros, Carla Gallo, Giovanni Poletti, Francisco Rothhammer, Resources, Data curation, Validation, Data acquisition, Final approval of the version to be published; Gabriel Bedoya, Resources, Data curation, Validation, Data acquisition, Funding acquisition, Final approval of the version to be published; André G Uitterlinden, Resources, Supervision, Funding acquisition, Project administration, Data acquisition, Final approval of the version to be published; M Arfan Ikram, Resources, Data curation, Supervision, Funding acquisition, Final approval of the version to be published; Eppo Wolvius, Steven A Kushner, Resources, Supervision, Final approval of the version to be published; Tamar EC Nijsten, Resources, Data curation, Supervision, Project administration, Data acquisition, Final approval of the version to be published; Robert-Jan TS Palstra, Conceptualization, Resources, Supervision, Methodology, Data acquisition, Final approval of the version to be published; Stefan Boehringer, Conceptualization, Resources, Software, Supervision, Methodology, Data acquisition, Final approval of the version to be published; Sarah E Medland, Resources, Supervision, Funding acquisition, Validation, Data acquisition, Final approval of the version to be published; Kun Tang, Data curation, Formal analysis, Funding acquisition, Validation, Investigation, Methodology, Data acquisition, Final approval of the version to be published; Andres Ruiz-Linares, Resources, Supervision, Funding acquisition, Validation, Project administration, Data acquisition, Manuscript preparation, Final approval of the version to be published; Nicholas G Martin, Conceptualization, Resources, Data curation, Supervision, Funding acquisition, Validation, Methodology, Project administration, Data acquisition, Manuscript preparation, Final approval of the version to be published; Timothy D Spector, Conceptualization, Resources, Data curation, Supervision, Funding acquisition, Methodology, Project

administration, Data acquisition, Manuscript preparation, Final approval of the version to be published; Evie Stergiakouli, Seth M Weinberg, Resources, Data curation, Supervision, Funding acquisition, Methodology, Project administration, Data acquisition, Manuscript preparation, Final approval of the version to be published; Fan Liu, Conceptualization, Resources, Software, Supervision, Funding acquisition, Visualization, Methodology, Data acquisition, Manuscript preparation and revision, Final approval of the version to be published; Manfred Kayser, Conceptualization, Resources, Supervision, Funding acquisition, Methodology, Project administration, Data acquisition, Manuscript preparation and revision, Final approval of the version to be published

### Author ORCIDs
Kaustubh Adhikari https://orcid.org/0000-0001-5825-4191
Mary L Marazita https://orcid.org/0000-0002-2648-2832
John R Shaffer https://orcid.org/0000-0003-1897-1131
Femke MS de Vrij https://orcid.org/0000-0003-0825-3806
Gemma C Sharp https://orcid.org/0000-0003-2906-4035
Carla Gallo https://orcid.org/0000-0001-8348-0473
Steven A Kushner https://orcid.org/0000-0002-9777-3338
Stefan Boehringer https://orcid.org/0000-0001-9108-9212
Manfred Kayser https://orcid.org/0000-0002-4958-847X

### Ethics
Human subjects: All cohort participants gave informed consent and consent to publish. The different cohort studies involved have been approved by their local ethics committees and in part higher institutions such as ministries, as described in the Materials and methods section. Protocol numbers can be found for each cohort in the Materials and methods section.

### Decision letter and Author response
Decision letter https://doi.org/10.7554/eLife.49898.sa1
Author response https://doi.org/10.7554/eLife.49898.sa2

## Additional files

### Supplementary files
• Supplementary file 1. Supplementary Tables. Includes: Supplementary Table 1. Characteristics of the all cohorts. Supplementary Table 2 Characteristics of 78 facial phenotypes in RS. Supplementary Table 3 Effects of 6 covariates to 78 facial phenotypes in RS. Supplementary Table 4. Phenotype correlations between all facial phenotypes in RS. Supplementary Table 5. Genetic correlation between all facial phenotypes in RS. Supplementary Table 6. Twins heritability and SNP-based heritability of all facial phenotypes in TwinsUK and QIMR. Supplementary Table7. 494 SNPs associated (p<5e-8) with facial shape phenotypes in meta-analysis of GWAS. Supplementary Table 8. Replication of reported variants of facial variation. Supplementary Table 9. GWAS Catalog entries (release on 2018-01-01), for polymorphisms that were significantly associated with facial traits in meta-analyses. Supplementary Table 10. Significant pleiotropic effect of study-wide suggestive SNPs on GeneATLAS traits. Supplementary Table 11. Frequency distribution of 24 top-associated SNPs. Supplementary Table 12. Population differentiation and positive selection at the suggestive significant GWAS SNPs. Supplementary Table 13. Replication of reported variants of NSCL/P trait. Supplementary Table 14. Multiple linear regression for 31 top-associated SNPs fitting with SEX and AGE in RS. Supplementary Table 15. Enrichment of the genes annotating the associated 24 regions. Supplementary Table 16. Significant expression quantitative trait loci effects for 494 SNPs significantly associated with facial traits and the target genes. Supplementary Table 17. Overviews of 5 selected candidate regulatory SNPs. Supplementary Table 18. Luciferase reporter assay results overview presented as relative luciferase activity fold change compared to construct containing less active allele ± standard deviation. Supplementary Table 19. Key Resource Table for functional experiments.

• Transparent reporting form

## Data availability

GWAS meta-analysis summary statistics data of the significantly associated SNPs are provided with the paper in the supplementary file 1. In addition, GWAS meta-analysis summary statistics of all SNPs and all facial phenotypes, including for each SNP the effect allele, non-effect allele and for each phenotype the effect size aligned to the effect allele with standard error and p-value, are made publicly available via figshare under https://doi.org/10.6084/m9.figshare.10298396 (updated file). Moreover, after the paper is accepted for publication, we will upload to the EBI GWAS Catalogue the complete summary statistics of all SNPs (same information as on figshare now) into the GWAS Catalogue. We included this information and the website links in the Material and Method section.

The following dataset was generated:

| Author(s) | Year | Dataset title | Dataset URL | Database and Identifier |
|---|---|---|---|---|
| Manfred Kayser | 2019 | GWAS facial shape variation in humans | https://doi.org/10.6084/m9.figshare.10298396 | figshare, 10.6084/m9.figshare.10298396 |

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
