## [Decision Letter]

**Acceptance summary:**

The reviewers appreciated the presentation of results of the meta-analysis of genome-wide association studies (GWAS) of 3D facial phenotypes in large sample sizes across cohorts of European ancestry, with replication in an additional three cohorts of diverse ancestry. The reviewers felt the detailed modelling of the facial phenotypes was a particular strength of the study. The authors identified 24 loci at stud-wide significance, 17 of which had not been previously reported: of these 10 loci were replicated (including six of the novel loci). Subsequent integration of these GWAS results with epigenomic data generated in cranial neural crest cells highlighted enrichment in cis-regulatory elements across the identified loci, and luciferase reporter assays demonstrated enhancer activity of several associated variants. Taken together, the reviewers felt that these data have substantially advanced the understanding of the genetic contribution to facial phenotypes, and have pinpointed potential causal genes and molecular mechanisms underlying facial variation that warrant further functional investigation.

**Decision letter after peer review:**

Thank you for submitting your article "Novel genetic loci affecting facial shape variation in humans" for consideration by *eLife*. Your article has been reviewed by three peer reviewers, including Andrew P Morris as the Reviewing Editor and Reviewer #1, and the evaluation has been overseen by Mark McCarthy as the Senior Editor. The following individual involved in review of your submission has agreed to reveal their identity: Seongwon Cha (Reviewer #3).

The reviewers have discussed the reviews with one another and the Reviewing Editor has drafted this decision to help you prepare a revised submission.

Summary:

Xiong et al. present results of meta-analysis of genome-wide association studies (GWAS) of facial phenotypes across cohorts of European ancestry, with replication in an additional three cohorts of diverse ancestry. The authors identified 24 loci at genome-wide significance, 17 of which had not been previously reported: of these 13 loci were replicated (including 9 of the novel loci). Integration of GWAS results with epigenomic data generated in cranial neural crest cells highlighted enrichment in cis-regulatory elements across the identified loci, and luciferase reporter assays demonstrated enhancer activity of variants in linkage disequilibrium with lead SNPs at three loci.

Essential revisions:

1) Genome-wide significance and replication. Given that 78 traits are tested here, using the traditional genome-wide significance threshold of 5x10^-8^ seems anti-conservative – some adjustment for multiple traits should be considered. This is even more essential as the replication is not ideal – only one of the three replication studies has exactly the same traits as discovery, and for the other two replication studies, a composite p-value for association with any facial trait is reported. The replication stage also does not take account of multiple testing of 24 SNPs. Only the composite p-value should be reported in replication. The authors could consider a meta-analysis of the p-value across the three replication studies using Fishers' method to give an overall assessment of replication evidence.

2) Elsewhere in the manuscript, "significant" is used, without any report of the threshold for significance (in the Results or the Materials and methods) or appropriate justification: "nominally significant" association with facial phenotypes; "significant" multi-tissue eQTL effects; no justification for p<0.05 in subsection “Preferential expression of face-associated genetic loci in embryonic cranial neural crest cells” and “Cis-regulatory signals in face-associated genetic loci”.

3) Signals of selection. More details of these findings should be reported in the Results section (and shorter comments in the Discussion). Since Fst and iHS are not well powered to detect selection, overlap with singleton density score (SDS) results in Europeans should be reported (Field et al., 2016).

4) Selection of "candidate regulatory SNPs" is not clear. The authors highlight SNPs that are in LD with the lead SNP at some loci. However, tag SNPs with r2 of 0.3 with the lead SNP do not seem to be in strong LD, and they may not actually show a strong association with the facial trait for that locus – as a result, the fact that the authors demonstrate enhancer activity for this SNP is irrelevant, as it will not be a good candidate to be driving the association signal. It also wasn't clear where the 5 SNPs came from, since 7 are reported in the preceding text. A better approach would be to define 99% credible sets of variants at each locus, which account for 99% of the probability of driving the association, and then assess the evidence for regulatory activity for these.

5) Multivariable modelling of phenotypes. The authors have considered all SNPs at genome-wide significance, and then LD pruned to a set of independent SNPs (r2<0.5). This does not define SNPs with independent effects on the phenotypes. The authors would be better to actually condition out the effects of the 24 lead SNPs by including them as covariates in the model, and then identify SNPs that remain at genome-wide significance. This could be done in exactly in RS, or could be done through approximate conditional analysis implemented in GCTA.

---

## [Author Response]

Essential revisions:1) Genome-wide significance and replication. Given that 78 traits are tested here, using the traditional genome-wide significance threshold of 5x10^-8^ seems anti-conservative – some adjustment for multiple traits should be considered. This is even more essential as the replication is not ideal – only one of the three replication studies has exactly the same traits as discovery, and for the other two replication studies, a composite p-value for association with any facial trait is reported. The replication stage also does not take account of multiple testing of 24 SNPs. Only the composite p-value should be reported in replication. The authors could consider a meta-analysis of the p-value across the three replication studies using Fishers' method to give an overall assessment of replication evidence.

We followed the reviewer’s suggestion, performed the requested analyses and revised our manuscript accordingly. In particular, a matrix spectral decomposition analysis of the total of 78 facial distance traits estimated the effective number of independent traits as 43, which corresponds to a Bonferroni corrected study-wide significant threshold of 1.2×10^-9^. We considered the traditional genome-wide significant threshold of 5×10^-8^ as our study-wide suggestive threshold for replication analysis. In the revised manuscript the study-wide significant and suggestive results were separately described. In addition, we applied the combined test method suggested by the reviewers to combine the evidence from the three replication datasets, and used the Bonferroni method to correct for the multiple testing of the 24 SNPs that passed the study-wide suggestive association threshold in the discovery analysis by applying the P<2.1x10^-3^ threshold. We now also applied Bonferroni correction to other analysis e.g. cis-qQTL, preferential expression, cis regulatory signals etc. The revised manuscript was adjusted accordingly.

2) Elsewhere in the manuscript, "significant" is used, without any report of the threshold for significance (in the Results or the Materials and methods) or appropriate justification: "nominally significant" association with facial phenotypes; "significant" multi-tissue eQTL effects; no justification for p<0.05 in subsection “Preferential expression of face-associated genetic loci in embryonic cranial neural crest cells” and “Cis-regulatory signals in face-associated genetic loci”.

As requested by the reviewer, we revised our association analysis using strict Bonferroni correction and reported the adjusted p-values. The revised manuscript was adjusted accordingly.

3) Signals of selection. More details of these findings should be reported in the Results section (and shorter comments in the Discussion). Since Fst and iHS are not well powered to detect selection, overlap with singleton density score (SDS) results in Europeans should be reported (Field et al., 2016).

As suggested by the reviewer, we looked-up our 24 face-associated SNPs in the External Online Resources of Field et al., 2016 and found that none of them overlapped with the previously reported signals of singleton density score (SDS, empirical p<0.01) in Europeans (Field et al., 2016). This may indicate that the observed selection signals are unlikely the consequences from the recent selective pressures as far as detectable via SDS analysis. We added this information to the revised manuscript.

4) Selection of "candidate regulatory SNPs" is not clear. The authors highlight SNPs that are in LD with the lead SNP at some loci. However, tag SNPs with r2 of 0.3 with the lead SNP do not seem to be in strong LD, and they may not actually show a strong association with the facial trait for that locus – as a result, the fact that the authors demonstrate enhancer activity for this SNP is irrelevant, as it will not be a good candidate to be driving the association signal. It also wasn't clear where the 5 SNPs came from, since 7 are reported in the preceding text. A better approach would be to define 99% credible sets of variants at each locus, which account for 99% of the probability of driving the association, and then assess the evidence for regulatory activity for these.

We agree with the reviewer that the description of this work was not clear in the submitted version. We made efforts to better clarify this in the revised version as follows. “In particular, 6 (25%) of the 24 face-associated loci, i.e., 1p36.13 *ARHGEF19,* 2q36.1 *PAX3, CMSS1*, 4q28.1 *INTU*, 16q12.1 *SALL1*, and 16q12.2 *RPGRIP1L*, showed significant enrichment of *cis*-regulatory entries at the top 0.1% of the genome level in at least one epigenomic tracks after Bonferroni correction (adjusted P<0.05, Figure 5—figure supplement 1). We selected 5 SNPs as candidate regulatory SNPs (Figures 5B, Supplementary file 1—table 17) as they not only overlapped with the putative enhancers defined in CNCCs (Prescott et al., 2015) and penis foreskin melanocyte primary cells (Chadwick, 2012; Consortium, 2012) (Figures 5A), but also showed high LD (*r^2^*>0.7) with at least one of the face-associated SNPs at the study-wide suggestive significance level. These were rs13410020 and rs2855266 from the 2q36.1 *PAX3* region, rs17210317 and rs6828035 from the 4q28.1 *INTU* region, and rs4783818 from the 16q12.2 *RPGRIP1L* region. These 5 candidate regulatory SNPs were further studied via functional experiments as described in the following section.” We believe that our approach is sufficiently systematic, while we also agree with the alternative approach suggested by the reviewer. However, because of the substantial amount of experimental work required to follow-up on the selected candidate SNPs, fully executing the reviewer’s approach including the necessary experimental follow-up exceeds the resources we currently have available. However, we will keep the reviewer’s suggestion in mind for future work.

5) Multivariable modelling of phenotypes. The authors have considered all SNPs at genome-wide significance, and then LD pruned to a set of independent SNPs (r2<0.5). This does not define SNPs with independent effects on the phenotypes. The authors would be better to actually condition out the effects of the 24 lead SNPs by including them as covariates in the model, and then identify SNPs that remain at genome-wide significance. This could be done in exactly in RS, or could be done through approximate conditional analysis implemented in GCTA.

Following the reviewer’s comment, we conducted a multiple-regression analysis in RS condition out the effects of the 24 lead SNPs by including them as covariates in the model. This analysis identified 31 SNPs showing significant (adjusted p<0.05) independent effects on facial traits, explaining up to 4.62% variance for individual phenotypes. The revised manuscript was adjusted accordingly. Performing approximate conditional analysis implemented in GCTA we do not feel comfortable because of the lack of individual-level data.